# The Hunchback transcription factor determines interneuron molecular identity, morphology, and presynapse targeting in the *Drosophila* NB5-2 lineage

Heather Q. Pollington, Chris Q. Doe[ID]*

Institute of Neuroscience, Howard Hughes Medical Institute, University of Oregon, Eugene, Oregon, United States of America

* cdoe@uoregon.edu

## Abstract

Interneuron diversity within the central nervous system (CNS) is essential for proper circuit assembly. Functional interneurons must integrate multiple features, including combinatorial transcription factor (TF) expression, axon/dendrite morphology, and connectivity to properly specify interneuronal identity. Yet, how these different interneuron properties are coordinately regulated remains unclear. Here we used the *Drosophila* neural progenitor, NB5-2, known to generate late-born interneurons in a proprioceptive circuit, to determine if the early-born temporal transcription factor (TTF), Hunchback (Hb), specifies early-born interneuron identity, including molecular profile, axon/dendrite morphology, presynapse targeting, and behavior. We found that prolonged Hb expression in NB5-2 increases the number of neurons expressing early-born TFs (Nervy, Nkx6, and Dbx) at the expense of late-born TFs (Runt and Zfh2); thus, Hb is sufficient to promote interneuron molecular identity. Hb is also sufficient to transform late-born neuronal morphology to early-born neuronal morphology. Furthermore, prolonged Hb promotes the relocation of late-born neuronal presynapses to early-born neuronal presynapse neuropil locations, consistent with a change in interneuron connectivity. Finally, we found that prolonged Hb expression led to defects in proprioceptive behavior, consistent with a failure to properly specify late-born interneurons in the proprioceptive circuit. We conclude that the Hb TTF is sufficient to specify multiple aspects of early-born interneuron identity, as well as disrupt late-born proprioceptive neuron function.

## Introduction

During development, the generation of neuronal diversity is a crucial component in assembling proper neural circuit assembly and increases the complexity of neural circuitry and behavioral output. The central nervous system (CNS) consists of a diverse array of neurons that differ in transcription factor (TF) and neurotransmitter expression, morphology, connectivity, and electrophysiology. All these features are integrated to define unique neuron identities, enabling each neuron to form specific and stereotyped neural circuits that generate appropriate behaviors.

**Data availability statement:** All relevant data are within the paper and its Supporting information files.

**Funding:** Funding was provided by HHMI (to CQD) and NIH HD27056 (to HQP). The funders had no role in study design, data collection and analysis, decision to publish, or preparation of the manuscript.

**Competing interests:** The authors have declared that no competing interests exist.

**Abbreviations:** CNS, central nervous system; GMC, ganglion mother cell; Grh, Grainy head; Hb, Hunchback; MDN, moonwalking descending neuron; NB, neuroblast; PN, projection neuron; RPC, retinal progenitor cell; TEM, transmission electron microscopy; TF, transcription factor; TTF, temporal transcription factor; VNC, ventral nerve cord.

Neurogenesis in both *Drosophila* and vertebrates is initiated by spatial cues that specify progenitor identity [1], followed by temporally expressed transcription factors (TTFs) that diversify the progeny of each progenitor [2–4]. In the mouse retina, retinal progenitor cells (RPCs) sequentially generate multiple cell types necessary for retinal circuit assembly [5]. RPCs express the TTF cascade: Ikaros> Pou2f1/Pou2f2> FoxN4> Casz1, each specifying specific cell types [4]. Ikaros expression is maintained in early-born RPC progeny and promotes expression of the homeodomain TF, Prox-1, required to specify early-born horizontal cell fate [6]. In the developing mammalian cortex, apical radial glia (aRG) produce cortical neurons that settle in functionally distinct laminar layers in an inside-out fashion, generating deep-layered early-born neurons and superficially located late-born neurons [3,4]. As in the retina, Ikaros is expressed early in neuronal progenitors of the mouse neocortex [7]. Prolonged Ikaros expression in cortical progenitors results in an increase in the number of early-born neurons expressing the early-born cortical TFs, Tbr1, and Foxp2, where it is required for early-born neuronal fate [7]. In the zebrafish spinal cord, early-born motor neurons and interneurons form a fast-swimming circuit, while later-born neurons form a slow-swimming circuit [8]. Early-born spinal cord neurons also show distinct morphological and connectivity differences compared to later-born neurons [8]. Thus, temporal patterning within progenitor lineages is an essential mechanism for generating neuronal diversity in the vertebrate CNS.

Similarly, *Drosophila* neural progenitors, called neuroblasts (NBs), sequentially produce motor neurons, interneurons, and glial cell type progeny that settle in a deep-to-superficial pattern [9–12], similar to mammalian cortical development [1]. *Drosophila* NBs generate a stereotyped sequence of neurons and glia in all parts of the CNS, but here we focus on the ventral nerve cord (VNC), analogous to the vertebrate spinal cord. NBs in the VNC sequentially express the following TTF cascade: Hunchback (Hb, ortholog of Ikaros)> Krüppel (Kr)> Nubbin/Pdm2 (Pdm)> Castor (Cas; ortholog of Casz1)> Grainy head (Grh) [2]. During each expression window, the NB asymmetrically divides to generate a smaller ganglion mother cell (GMC), which then divides to produce two post-mitotic neurons that transiently maintain the TTF expressed at the time of birth. This generates early-born neurons that express Hb, required for establishing motor neuron identity [9,13–17].

The role of Hb in specifying *Drosophila* motor neuron fate has been most extensively investigated in the NB3-1 and NB7-1 lineages [9,13–17]. NB7-1 generates the U1-U5 and VO motor neurons in sequential birth order: U1/U2 motor neurons express Hb, U3 expresses Kr, VO expresses Kr/Pdm, U4 expresses Pdm, and U5 expresses Pdm/Cas [9]. Prolonged Hb expression in NB7-1 generates increased numbers of U1/U2 at the expense of later-born motor neurons [9,14,15,18]. The late-born ectopic U1/U2 motor neurons lack expression of the late-born TF, Zfh2, display U1/U2 morphology, and form synapses with U1/U2 dorsal muscle targets [14,15]. Similarly, prolonged Hb expression in NB3-1 increases functional synapses innervating muscles targeted by early-born neurons and decreases synapse formation to muscles innervated by later-born neurons [16]. Increased synaptic connections to early-born muscles increase presynaptic vesicle release; however, there is no difference in miniature-excitatory post-synaptic potential amplitude, suggesting ectopic Hb motor neurons may undergo homeostatic compensation [16].

The role of Hb in generating motor neuron identity and connectivity is well characterized, yet relatively little is known about the role of TTFs in generating interneuron identity. Interneuron and motor neuron axons have different constraints on circuit formation. While motor neuron axons select a muscle target from a small pool of muscles, interneuronal axons must bypass a vast array of potential neuron partners within the synaptically dense neuropil, a much more complex environment, to form appropriate synaptic connections to partner neurons. In addition, interneuron neurites are directed by gradients of the attractant and

repulsive extrinsic signaling cues: Semaphorin 1a, 2a, and 2b, Slit, and Netrin [19–23]. Interneurons expressing specific guidance receptors are either attracted or repelled from discrete domains within the neuropil, directing neurites to extend to distinct neuropil regions. These extrinsic cues may contribute to interneuron morphology and connectivity; however, the role of neuron intrinsic mechanisms is not well understood.

To understand whether the mechanisms that Hb uses to specify early-born motor neuron identity are conserved in interneurons, we investigated the role of Hb in specifying interneuron identity in the NB5-2 lineage. NB5-2 predominately generates interneuron progeny, with the late-born interneurons forming a proprioceptive circuit [11,24]. We generated a NB5-2 split Gal4 line to genetically label NB5-2 and its progeny. We found that NB5-2 expresses the canonical TTF cascade and TTF expression is transiently maintained in the interneuronal progeny. When we prolonged expression of Hb in the NB5-2 lineage, it resulted in an increased number of Hb+ NB5-2 interneurons that express early-born TFs (Nervy, Dbx, and Nkx6) at the expense of neurons expressing late-born TFs (Runt and Zfh2). In addition, we identified several early-born Hb+ neurons with an axon/dendrite morphology distinct from that of later-born interneurons, forming a unique diagonal contralateral projection. At the synaptic level, ectopic Hb expression in late-born neurons resulted in an increase in presynapse percentage at neuropil locations normally targeted by NB5-2 Hb+ early-born neuronal presynapses. Finally, we found that prolonged NB5-2 Hb expression generates behavioral defects similar to disrupting the late-born proprioceptive circuit: decreased locomotor velocity and increased number of C-shaped body bends. In summary, we show that Hb specifies early-born NB5-2 interneuron molecular identity, axon/dendrite morphology, and presynapse targeting (essential for proper connectivity) and that proper TTF identity is necessary for appropriate behavioral output.

## Results

### Generation of a split-Gal4 line expressed in NB5-2 and its embryonic lineage

To create a transgene specifically expressed in the NB5-2 lineage, we constructed a split-Gal4 based on the intersection of *wingless* (*wg*), expressed in row 5 neuroectoderm, and *ventral nervous system defective* (*vnd*), expressed in medial column neuroectoderm (Fig 1A and 1B). The intersection of these two hemi-drivers defines the neuroectoderm domain that generates NB5-2, and the split-Gal4 is strongly and specifically expressed in NB5-2 and its progeny (Fig 1C). We refer to this split-Gal4 line alone as NB5-2-Gal4, or NB5-2 > GFP when the NB5-2-Gal4 is driving expression of UAS-GFP. We note that NB5-2-Gal4 has occasional expression in the NB5-1 lineage, but this has no effect on our experiments as NB5-1 delaminates much later in embryogenesis, starts its lineage Cas+, and thus is unlikely to express earlier TTFs Hb, Kr, and Pdm [25]. Thus, we conclude that NB5-2-Gal4 is an appropriate tool for driving gene expression in NB5-2 and its embryonic progeny.

### NB5-2 expresses the canonical TTF cascade

Most VNC NB lineages sequentially express Hb, Kr, Pdm, Cas, and Grh and include intervals where adjacent TTFs are co-expressed [2]. Here we determine whether NB5-2 also expresses this TTF cascade, and whether there are gene expression overlaps within the series. We found that NB5-2 > GFP expression was not detected until late stage 10, and thus at earlier stages NB5-2 was identified by its position in the NB array: the most medial row 5 NB just anterior to the row 6/7 Engrailed expression domain (S1 Fig). Using these markers prior to stage 10, and NB5-2 > GFP expression beginning at stage 10, we were able to characterize the

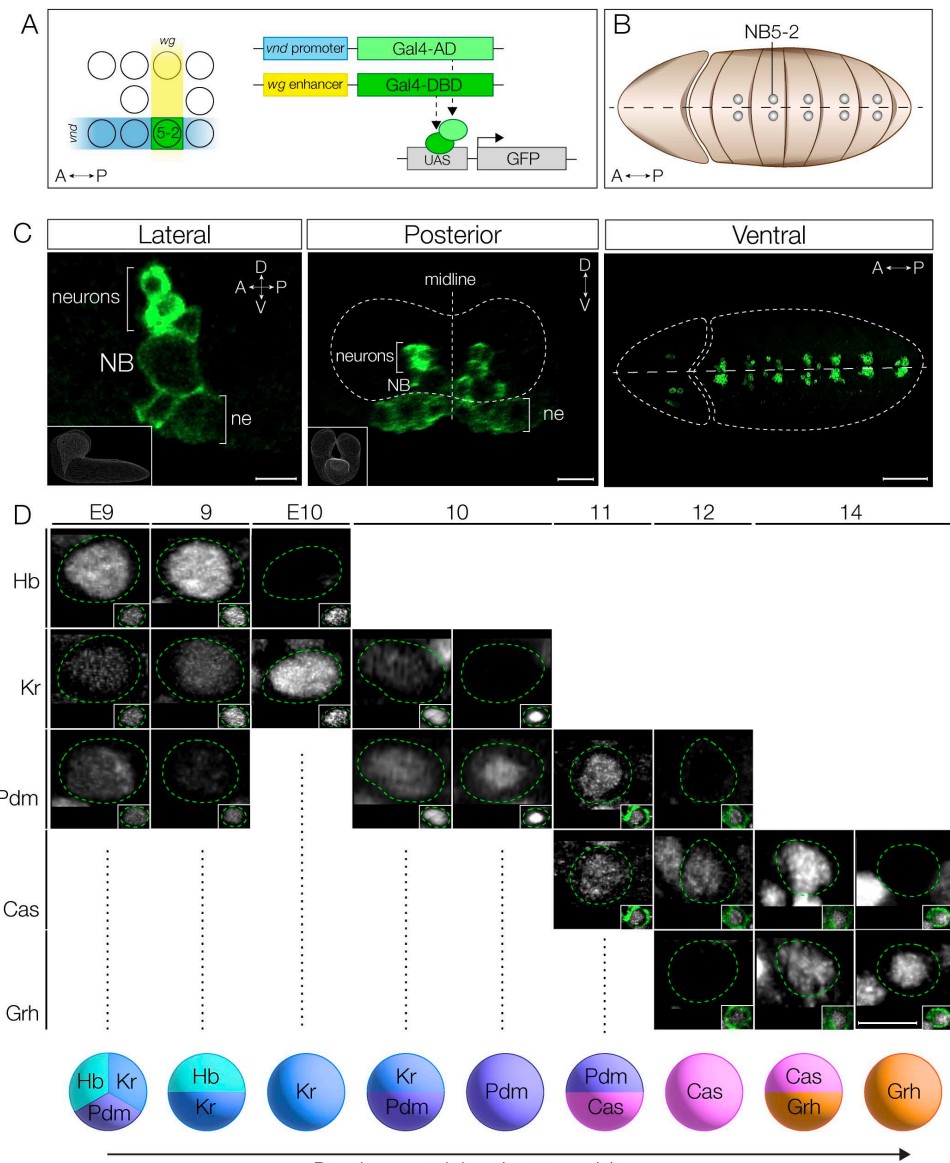

**Fig 1. NB5-2 expresses the canonical TTF cascade.** (**A**) Schematic of NB5-2 split Gal4 (green) construction using the unique NB5-2 expression combination of *wingless* (*wg*; yellow) and *ventral nervous system defective* (*vnd*; blue). (**B**) Schematic of segmentally repeating NB5-2 location in early embryo. Anterior left, ventral view. (**C**) NB5-2 > GFP expression in stage 11 embryo in the lateral view (left panel; inset shows CNS volume from lateral view; Scale bar: 5 μm), posterior view (middle panel; inset shows CNS volume from posterior view; Scale bar: 10 μm), and ventral view (right panel; Scale bar: 50 μm) labels NB5-2, NB5-2 neuron progeny, and a small population of neuroepithelium (ne). (**D**) NB5-2 TTF expression cascade at stages 9–14. NB5-2 identification was achieved using the NB markers Deadpan (Dpn; insets in all expect stg 10) or Worniu (Wor; inset stg 10 only). Due to delayed GFP (green) expression in stage 9 and early stage 10 embryos, NB5-2 was identified as the most medially located NB, anteriorly adjacent to the Engrailed expression domain (see S1 Fig). A GFP labeled NB5-2 was used to identify NB5-2 after early stage 10. Quantification: Stg E9 *n* = 16 hemisegments, 4 animals; stg 9 *n* = 12 hemisegments, 3 animals; stg E10 *n* = 20 hemisegments, 5 animals; stg 10 *n* = 12 hemisegments, 3 animals; stg 11 *n* = 8 hemisegments, 2 animals; stg 12 *n* = 12 hemisegments, 3 animals; stg 14 *n* = 12 hemisegments, 3 animals. Scale bar: 5 μm.

TTF cascade in NB5-2 throughout embryogenesis (Fig 1D). We found that NB5-2 initially expresses Hb, low levels of Kr, and transient Pdm; brief expression of Pdm during the NB first division window has been previously observed in the NB4-2 lineage and thought to be protein inherited from the Pdm+ neuroectoderm [26]. Subsequently, NB5-2 sequentially expresses Kr, Pdm, Cas, and Grh with a period of overlap in each case (Figs 1D and S1). We conclude that NB5-2 undergoes the canonical TTF cascade, with transient overlap of each TTF, resulting in nine different TTF expression windows. Most importantly for this work, Hb is detected in the earliest portion of the NB5-2 lineage.

## NB5-2 neuronal progeny express the TTF present at their time of birth

In most NB lineages, TTF expression in the NB is transiently maintained in the progeny born during each NB TTF expression window [9,12]. Indeed, we find that each TTF can be detected within a subset of NB progeny (Fig 2A–2C; quantified in 2D). Each window of NB expression, both single and dual gene overlap, contributes to the neuronal progeny (Fig 2A–2C; quantified in 2D). Transient expression varied among TTFs: Hb, Kr, and Grh expression was maintained until the L1 larval stage (and possibly longer); Pdm expression was detectable only until embryonic stage 14; and Cas was detectable until late embryonic stage 17, with little expression observed in newly hatched larvae. Neurons expressing early TTFs are located in a deep layer closer to the neuropil, while neurons expressing later TTFs are located more superficially (Fig 2A–2C), as expected from previous work [9,11,12]. In addition, neurons from sequential TTF windows invariably have adjacent cell body positions (Fig 2A–2C).

Interestingly, there are different numbers of neurons expressing each TTF (or TTF combination) (Fig 2D), which allows us to use neuron numbers as a proxy for the length of each TTF expression window. For example, at stage 14, Hb is detected in 5 neurons, which suggests that the NB Hb expression persisted for three divisions (each division making a Hb+ GMC that makes a pair of sibling neurons). Although we note that NB5–2-Gal4 does not begin expression until after the first NB division in some animals, thus marking only 3–4 of the 5 Hb+ neurons. In this way, we infer that Kr alone has a one division window to contribute 2 neurons, Kr/Pdm has one division window (2 neurons), Pdm alone has a one division window (2 neurons), Pdm/Cas has a two division window (4 neurons), Cas alone has a two division window (4 neurons), Cas/Grh has a one division window (2 neurons), and at stage 16, Grh alone has a one division window (2 neurons) (summarized in Fig 2E). We also note that the average total number of NB5-2 neurons at stage 14 compared to stage 16/17 is consistent with the generation of Cas/Grh and Grh alone neurons after stage 14 (Fig 2F).

## Neuronal settling position is correlated with birth order

Previous work has shown that early-born Hb+ neurons reside closest to the neuropil, being physically displaced into more dorsal regions by later-born neurons. Conversely, later-born Cas+ neurons take more superficial positions [9–12]. To determine if NB5-2 progeny settling position is correlated with birth order, we mapped the deep-to-superficial position of the NB5-2 progeny. We found that Hb+ and Kr+ neurons are located in overlapping deep layers (Fig 3A). Pdm neurons are positioned more superficially (Fig 3A–3B), with some overlap with both Hb and Kr, consistent with overlapping expression patterns seen in NB5-2 (Fig 1). Lastly, both Cas + and Grh + neuron populations are located in the most superficial layers (Fig 3B–3C). We conclude that (1) neuronal settling position is correlated with birth order in the NB5-2 lineage (Fig 3D), and (2) that TTF expression, neuronal settling position, and birth order are tightly correlated. These findings all support a model in which each TTF is inherited by the neuronal progeny born during each TTF expression window, with little migration

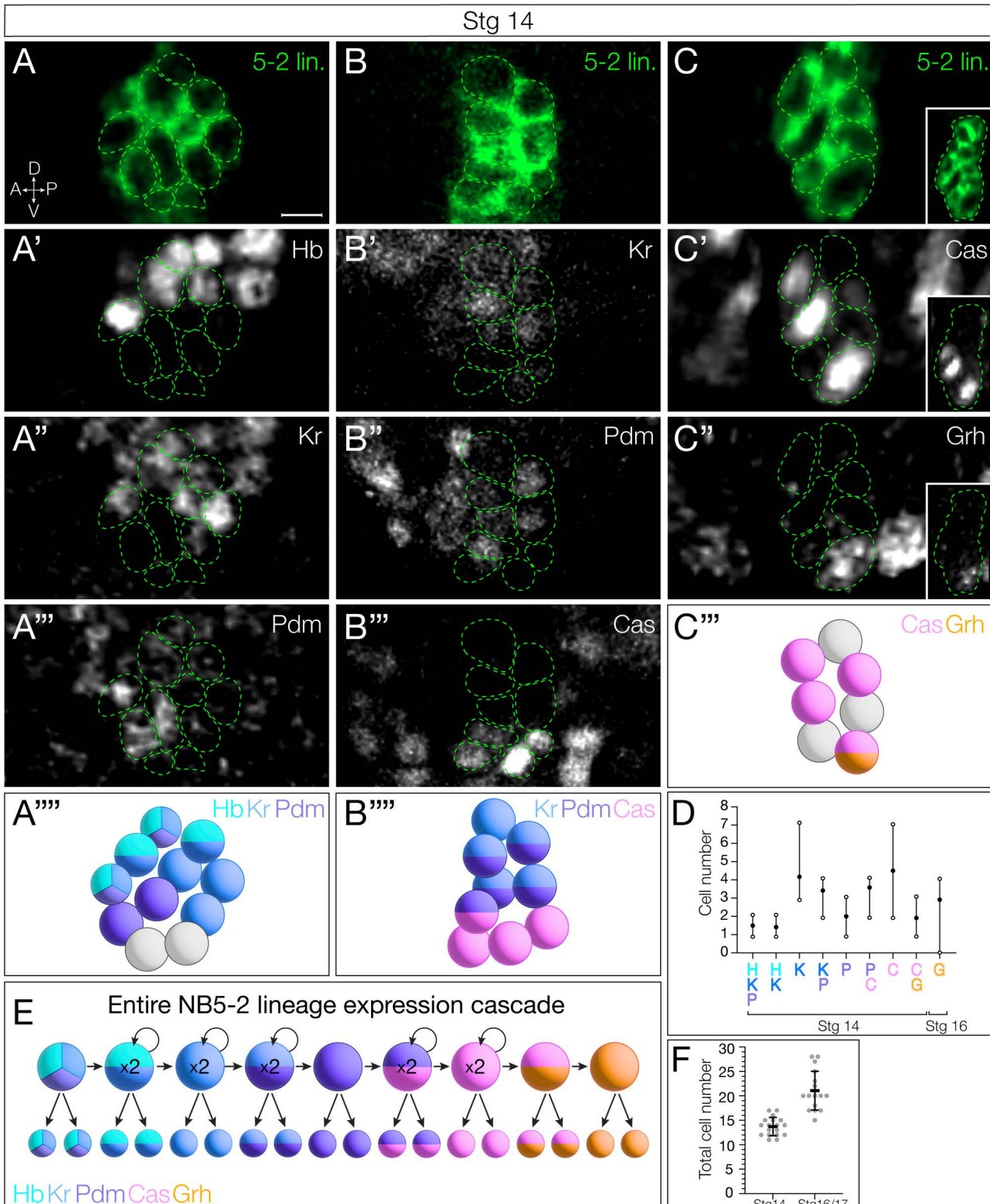

**Fig 2. NB5-2 progeny maintain TTF expression present at time of birth.** (**A**) NB5-2 > GFP progeny (green) stained for Hb (**A′**), Kr (**A″**), and Pdm (**A‴**) at stage 14; A schematic summary (**A″″**). Anterior left, lateral view. Scale bar: 4 μm. (**B**) NB5-2 > GFP progeny stained for Kr (**B′**), Pdm (**B″**), and Cas (**B‴**); schematic summary (**B″″**). (**C**) NB5-2 > GFP progeny stained for Cas (**C′**) and Grh (**C″**) with a low magnification image of Cas-/Grh- neurons (inset); schematic summary (**C‴**). (**D**) Quantification of NB5-2 neurons expressing each TTF singly or in combination in stage 14 (all but Grh)

and 16 embryos (only Grh). Hb/Kr/Pdm avg = 1.5, Hb/Kr avg = 1.4, Kr avg = 4.2, Kr/Pdm avg = 3.4, Pdm avg = 2, Pdm/Cas = 3.6, Cas avg = 4.5, Cas/Grh avg = 1.9, Grh avg = 2.9; $n$ = 12 hemisegments, 3 animals. (**E**) Schematic of NB5-2 divisions during each TTF expression window throughout embryogenesis. (**F**) Quantification of total NB5-2 neurons in stage 14 embryos (avg = 13.9, $n$ = 18 hemisegments, 5 animals) and 16 (avg = 21.1, $n$ = 17 hemisegments, 5 animals). The data underlying the graphs in the figure can be found in S1 Data.

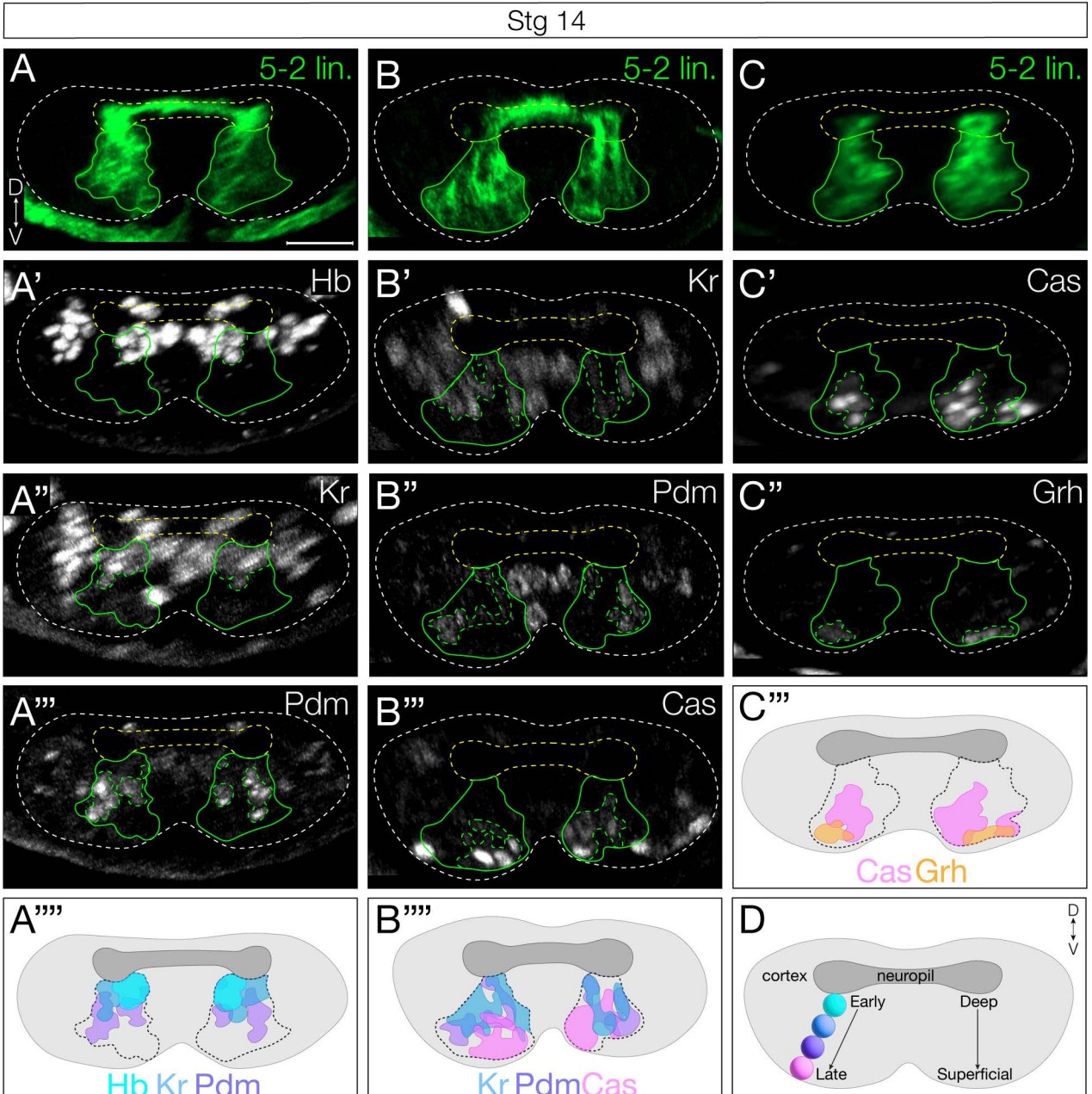

**Fig 3. NB5-2 progeny deep-superficial settling position is correlated with birth order.** (**A**) NB5-2 > GFP progeny (green) settling position of neurons expressing Hb (**A′**), Kr (**A″**), and Pdm (**A‴**); schematic summary (**A″″**). Dorsal up, posterior view. $n$ = 6 segments, 3 animals. Scale bar: 10 µm. (**B**) NB5-2 > GFP progeny settling position of neurons expressing Kr (**B′**), Pdm (**B″**), and Cas (**B‴**); schematic summary (**B″″**). (**C**) NB5-2 > GFP progeny settling position of neurons expressing Cas (**C′**) and Grh (**C″**); schematic summary (**C‴**). (**D**) Cartoon of stereotypical early-to-late born cell body positions within the VNC cortex.

or movement, resulting in early-born neurons located in deep layers and late-born neurons located in superficial layers.

## Prolonged Hunchback expression in NB5-2 prevents late TTF expression

To distinguish the relative importance of birth order and TTF expression in specifying interneuronal identity, we broke their correlation by mis-expressing Hb in the NB5-2 lineage, which allowed us to determine which—birth order or TTF identity—is more important for neuronal identity. We used NB5-2-Gal4 to specifically misexpress Hb throughout the lineage, and assay for alterations in the expression of later-born TTFs. If time of birth is used to generate interneuron diversity, we would expect little effect of Hb misexpression; in contrast, if the TTF identity is causal for interneuron identity, we would expect to see a clear transformation of late-born to early-born interneuron identity. In the following sections, we determine the effect of Hb misexpression on interneuronal molecular identity, axon/dendrite morphology, and presynapse targeting within the dense neuropil.

We first validated the ability of Hb misexpression to repress later NB TTFs; in other NB lineages, misexpression of Hb can result in delayed or absent expression of late TTFs [9]. The timing of Hb misexpression is also important: NBs have a relatively short competence window for responding to Hb [27,28]. We found that NB5-2-Gal4 drives misexpression of Hb beginning at early stage 10, soon after the termination of endogenous Hb expression (Fig 4A), and prior to the termination of the competence window at stage 12 [27]. Importantly, we found that misexpression of Hb in the NB5-2 lineage resulted in a highly penetrant loss of expression of all later-born TTFs (Fig 4A; summarized in 4B), a prerequisite for subsequent experiments.

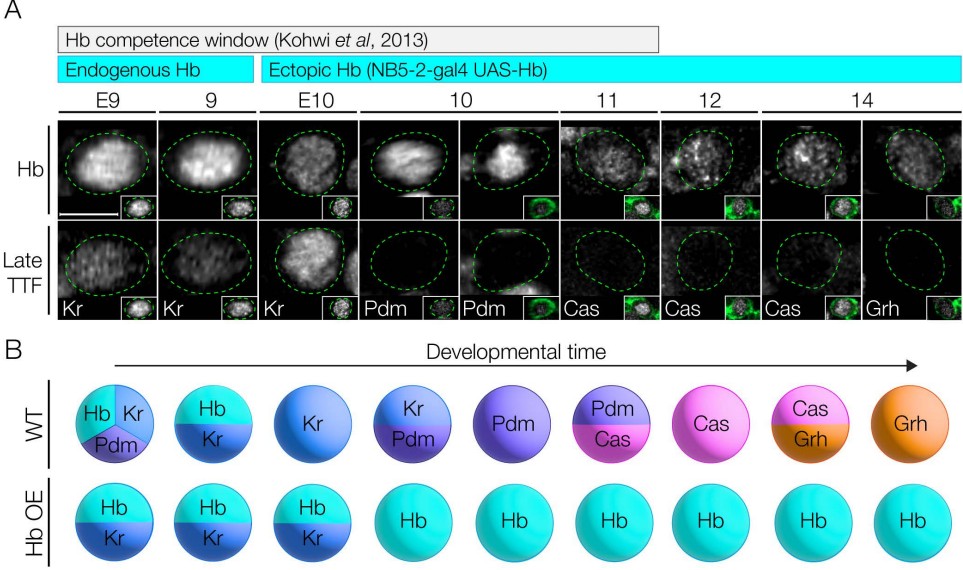

**Fig 4. Prolonged Hunchback expression in NB5-2 delays expression of late TTFs.** (**A**) NB5-2 > Hb progeny expressing Hb (top panels) and later TTFs (bottom panels) at stages 9–14. Bars show developmental length of known Hb competence window in other NBs (gray) and endogenous and ectopic NB5-2 Hb expression (cyan). NB5-2 identification was achieved as in S1 Fig; a combination of GFP (green) and Dpn was used to identify NB5-2 after stage 10 (insets). Quantification: stg E9 $n$ = 20 hemisegments, 5 animals; stg E10 $n$ = 24 hemisegments, 6 animals; stg 10 $n$ = 16 hemisegments, 4 animals; stg 11 $n$ = 8 hemisegments, 2 animals; stg 12 $n$ = 12 hemisegments, 3 animals; stg 14 $n$ = 12 hemisegments, 3 animals. Scale bar: 5 μm. (**B**) Schematic of wildtype and Hb overexpression in NB5-2.

## Hunchback specifies early-born interneuron molecular identity

Next, we wanted to determine if loss of later-born TTFs due to prolonged Hb expression extended to NB5-2 progeny. We quantified the number of NB5-2 neurons expressing each TTF following Hb misexpression. As expected, we observed a significant increase in the number of Hb+ neurons in the NB5-2 lineage (Fig 5E and 5F, quantified in 5Q). We also found an increase in Hb+/Kr+, and Hb/+Pdm+ co-expressing neurons (S2A and S2B Fig, quantified in S2D–S2G), consistent with co-expression observed in wildtype NB5-2 neurons (Fig 2A). We note that while NB Pdm expression is repressed by Hb misexpression, we see an increase in Hb+/Pdm+ NB5-2 progeny; the discrepancy in Pdm expression from NB to progeny is unclear. Conversely, we observed a striking decrease in Cas+ and Grh+ neurons, of which NB5-2 Hb+ neurons have no overlapping expression (Fig 5K and 5L, quantified in 5U; S2C Fig, quantified in 5H). We conclude that NB5-2 > Hb is an effective tool for prolonging Hb expression throughout the NB5-2 lineage, and that prolonged Hb expression is highly effective at repressing NB expression of all later TTFs.

To determine the role of Hb in specifying interneuron molecular identity downstream of TTF expression, we needed to identify molecular markers for early-born Hb+ and late-born Hb-negative interneurons in the NB5-2 lineage. We screened a collection of TF antibodies, and identified Nervy, Dbx, and Nkx6 as early-born TFs expressed in Hb+ neurons but not in Cas+ neurons (Figs 5A, 5B, and S3A). Conversely, we identified Runt and Zfh2 as late-born TFs expressed in Cas+ neurons, but not in Hb+ neurons (Fig 5C and 5D). Next, we prolonged Hb expression in the NB5-2 lineage and assayed for changes in TF expression. We found that Hb misexpression resulted in an expansion of early-born Hb+, Nervy+, Dbx+, and Nkx6+ neurons that spanned the deep-superficial axis of the NB5-2 lineage (Figs 5E–5J, S3B, and S3C; quantified in 5Q–5T). Furthermore, prolonged Hb expression resulted in a striking loss of late-born Cas+, Runt+, and Zfh2+ neurons (Fig 5K–5P; quantified in 5U–5W). Importantly, there was no significant difference in the total number of NB5-2 neurons per hemisegments (Fig 5X), showing that late-born neurons are transformed into neurons with an early-born molecular identity, rather than undergoing cell death. We conclude that Hb is sufficient to specify early-born interneuron molecular identity at the expense of late-born neurons, and that temporal identity is more important than birth order in specifying early-born interneuron molecular identity.

## Hunchback+ early-born interneurons have a distinct morphology compared to later-born interneurons

To determine the role of Hb in specifying NB5-2 early-born interneuron axon/dendrite morphology, we needed to identify the individual neuron morphology of both early-born Hb+ and late-born Hb-negative interneurons. We showed above that NB5-2 makes 5 Hb+ neurons (Fig 2). Here, we design a genetic method for identifying the morphology of each Hb+ neuron in the NB5-2 lineage. We used intersectional genetics to stochastically label individual neurons that were both Hb+ and derived from NB5-2. We call this intersectional approach "Hb+ filtered NB5-2 neurons" (Fig 6A). Using this method, we identified 5 Hb+ neurons in the NB5-2 lineage: MN12 (Fig 6B), and 4 interneurons that we name "Idun1-Idun4" (Fig 6C–6F). Each Idun neuron has a similar but unique morphology. Based on deep-superficial cell body position, we propose that the first NB5-2 derived GMC (GMC-1) makes MN12/Idun1 siblings, GMC-2 makes Idun2/Idun3 siblings, and GMC-3 makes Idun4 and a sibling that may undergo apoptosis, as we have never detected a sixth Hb+ neuron in the lineage. Subsequently, we focus on the four Hb+ Idun interneurons.

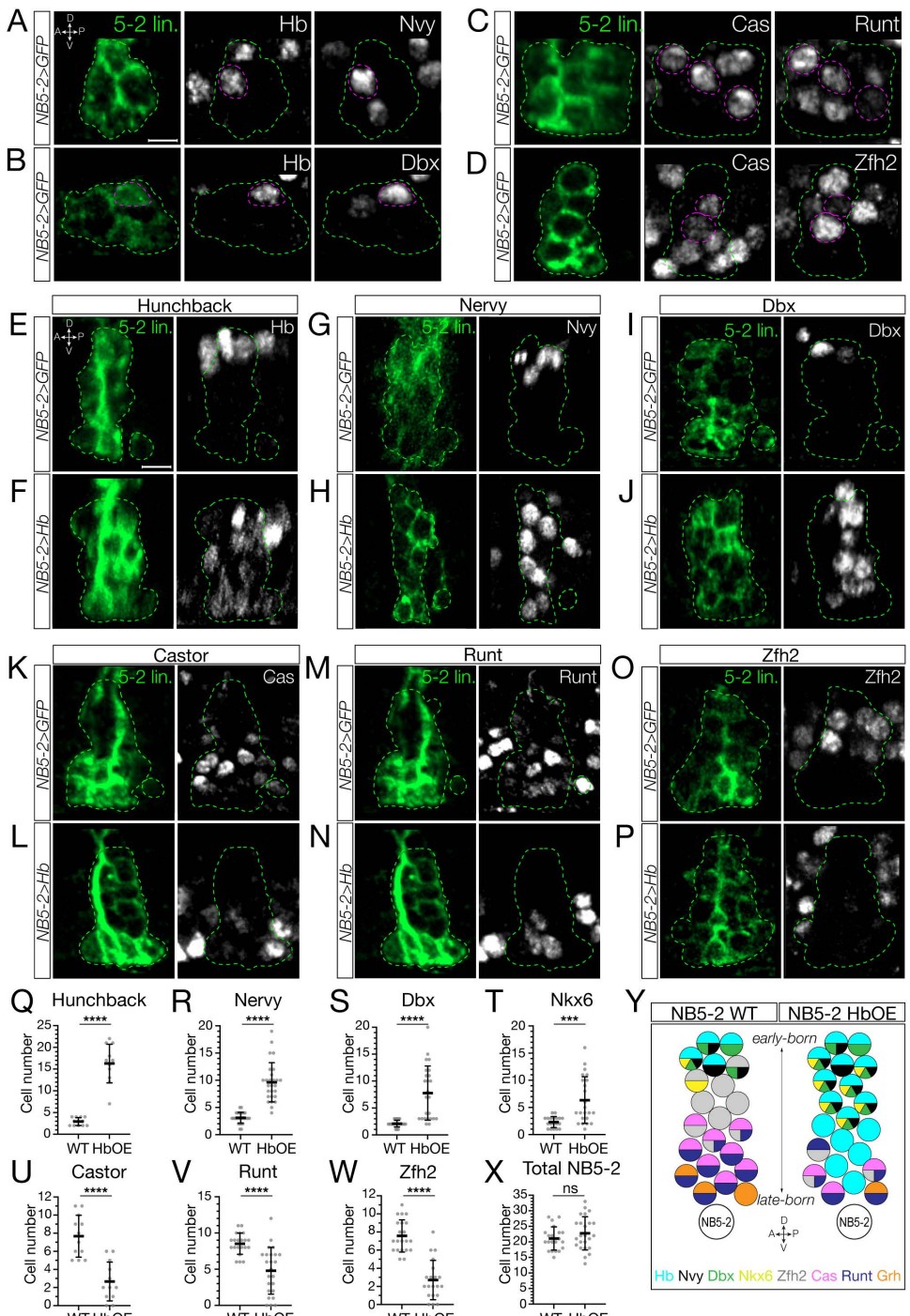

**Fig 5. Prolonged Hunchback expression in NB5-2 generates ectopic early-born interneurons at the expense of late-born neurons.** (**A, C, E, G, I, K, M, O**) Wildtype NB5-2 > GFP progeny (green; dashed outline) stained for the indicated markers. Stage 16/17 (unless stated otherwise), anterior left, lateral view. Scale bar: 4 μm. (**B, D, F, H, J, L, N, P**) NB5-2 > Hb (green; dashed outline) stained for the indicated markers. Stage 17, anterior left, lateral view. Scale bar: 4 μm. (**Q–X**) Quantification. The data underlying the graphs in the figure can be found in S2 Data. (**Q**) Hb neurons. Stage 14. WT avg = 2.75, $n = 12$ hemisegments, 3 animals; HbOE avg = 16.27, $n = 11$ hemisegments, 3 animals. (**R**) Nervy neurons. WT NB5-2 avg = 3.11, $n = 28$ hemisegments, 7 animals; HbOE avg = 9.63, $n = 27$ hemisegments, 7 animals. (**S**) Dbx neurons. WT NB5-2 avg = 2.09, $n = 22$ hemisegments, 6 animals; HbOE avg = 7.76, $n = 25$ hemisegments, 7 animals. (**T**) Nkx6 neurons. WT NB5-2 avg = 2.31, $n = 19$ hemisegments, 5 animals; HbOE avg = 6.35, $n = 23$ hemisegments, 6 animals. (**U**) Castor neurons. WT NB5-2 avg = 7.67, $n = 12$ hemisegments, 3 animals; HbOE

avg = 2.67, $n$ = 12 hemisegments, 3 animals. (**V**) Runt neurons. WT NB5-2 avg = 8.52, $n$ = 21 hemisegments, 6 animals; HbOE NB5-2 avg = 4.79, $n$ = 24 hemisegments, 6 animals. (**W**) Zfh2 neurons. WT NB5-2 avg = 7.57, $n$ = 21 hemisegments, 6 animals; HbOE NB5-2 avg = 2.7, $n$ = 20 hemisegments, 6 animals. (**X**) Total NB5-2 progeny number. WT avg = 21.10, $n$ = 21 hemisegments, 6 animals; HbOE avg = 22.75, $n$ = 24 hemisegments, 6 animals. (**Y**) Schematic summary.

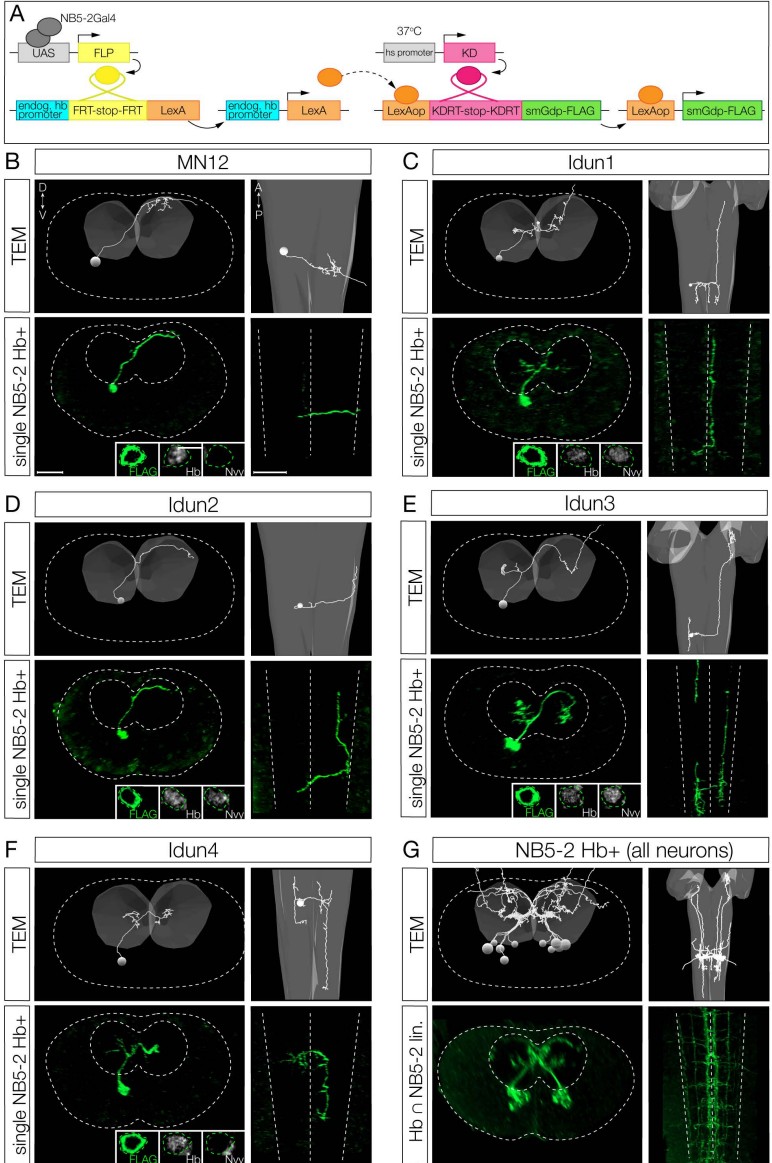

**Fig 6. Hunchback+ early-born interneurons have a distinct morphology compared to later-born neurons.** (**A**) Intersectional genetics was used to stochastically label NB5-2 Hb+ neurons. See methods for details. (**B–F**) Single-cell morphology for the indicated neurons in L1 (0–3 h ALH) larvae. Top panels, TEM reconstruction; left, posterior view; right, dorsal view. Dotted line marks VNC border; gray surface, neuropil. Bottom panels, single neuron labeling of the indicated neurons using Hb+ filtered genetics. Left, posterior view, dotted lines mark the VNC and the neuropil, inset shows Hb expression in each neuron; right, dorsal view; dotted lines indicate the VNC borders and the midline. (**G**) TEM reconstruction (top panels) and Hb+ NB5-2 lineage labeling (bottom panels) of all Hb+ NB5-2 neurons.

We previously used the open-source transmission electron microscopy (TEM) volume of the first instar larval CNS [29] to reconstruct all of the neurons within the NB5-2 lineage in both segment A1L and A1R [11], but it was unknown which neurons in the whole lineage reconstruction were Hb+. To identify the morphology of the Hb+ neurons in the lineage, we used the intersectional genetics method described above to label "Hb+ filtered NB5-2 neurons" with a membrane-bound reporter (Fig 6A). We were able to identify all four Hb+ Idun interneurons as well as MN12 based on a clear match in neuron morphology (Fig 6B–6F). All Hb+ Idun interneurons are unique, but there are a few shared features. (1) All Idun interneurons have a cell body position in a deep layer close to the neuropil. (2) All Idun interneurons have an ascending or descending intersegmental projection. (3) With the exception of Idun4, all Idun interneurons have a contralateral projection that crosses the midline with a diagonal trajectory, from ventral-medial to dorso-lateral, clearly visible in a posterior view.

We found that the formation of a diagonal contralateral projection was unique to the early-born neurons; TEM reconstruction of all neurons in the lineage only found this trajectory in Idun1, Idun2, Idun3, and MN12. These shared morphological features can be observed when all five Hb+ neurons in the lineage are labeled together by light or electron microscopy (Fig 6G). In conclusion, we have identified four Hb+ interneurons in the NB5-2 lineage and matched them to the same four interneurons within the TEM reconstruction. Importantly, these early-born interneuron morphologies are clearly different from late-born interneuron morphology (S4 Fig). In addition to identifying morphological differences in early-born and late-born interneurons, mapping the Idun interneurons in the TEM volume allows us to quantify presynapse and postsynapse position for each Idun neuron—a prerequisite for characterization of Idun presynapse neuropil targeting (see below).

## Hunchback specifies early-born interneuron morphology

We showed above that Hb determines early-born interneuron molecular identity based on early and late-born TF expression. Here, we ask whether Hb determines early-born interneuron axon/dendrite morphology. We wanted to distinguish between two possible mechanisms, as we did for interneuron molecular identity. First, *birth-order* may be critical for determining neuronal morphology, perhaps each neuron born at a different time encounters a changing environment that directs the appropriate projection. Second, *TTF expression* may determine neuronal morphology, perhaps via TTF-regulated guidance cues. To determine the relative importance of birth order and TTF expression, we misexpressed Hb in the NB5-2 lineage and asked whether late-born neurons persisted in generating late-born neuronal morphology (birth-order model) or whether they acquired early-born neuronal morphology (TTF model). The latter was found to be important in the motor neurons of the NB7-1 lineage [14,15]; here, we ask whether this mechanism is extended to the interneurons, where axons and dendrites remain within the synaptically dense neuropil of the CNS.

We focus on Idun1, Idun2, and Idun3, as these interneurons have a unique diagonal commissural crossing projection not observed in later-born neurons, with the addition of Idun1 possessing the most medial ascending projection and Idun2 possessing the most lateral ascending projection among NB5-2 progeny (Fig 7). NB5-2 > GFP labels three types of contralateral projections: dorsal, ventral, and diagonal, that are best seen in a posterior view (Fig 7A), plus ascending/descending projections that are best seen in a dorsal view (Fig 7A′). In contrast to controls, Hb misexpression shows a striking loss of the dorsal and ventral contralateral projections and an increase in the diagonal projections (Fig 7B; posterior view) as well as a loss of ascending/descending projections (Fig 7B′). These phenotypes are due to a change in late-born neuron morphology, and not the death of late-born neurons, as there are the same number of neurons in control and experimental NB5-2 lineages (Fig 5X).

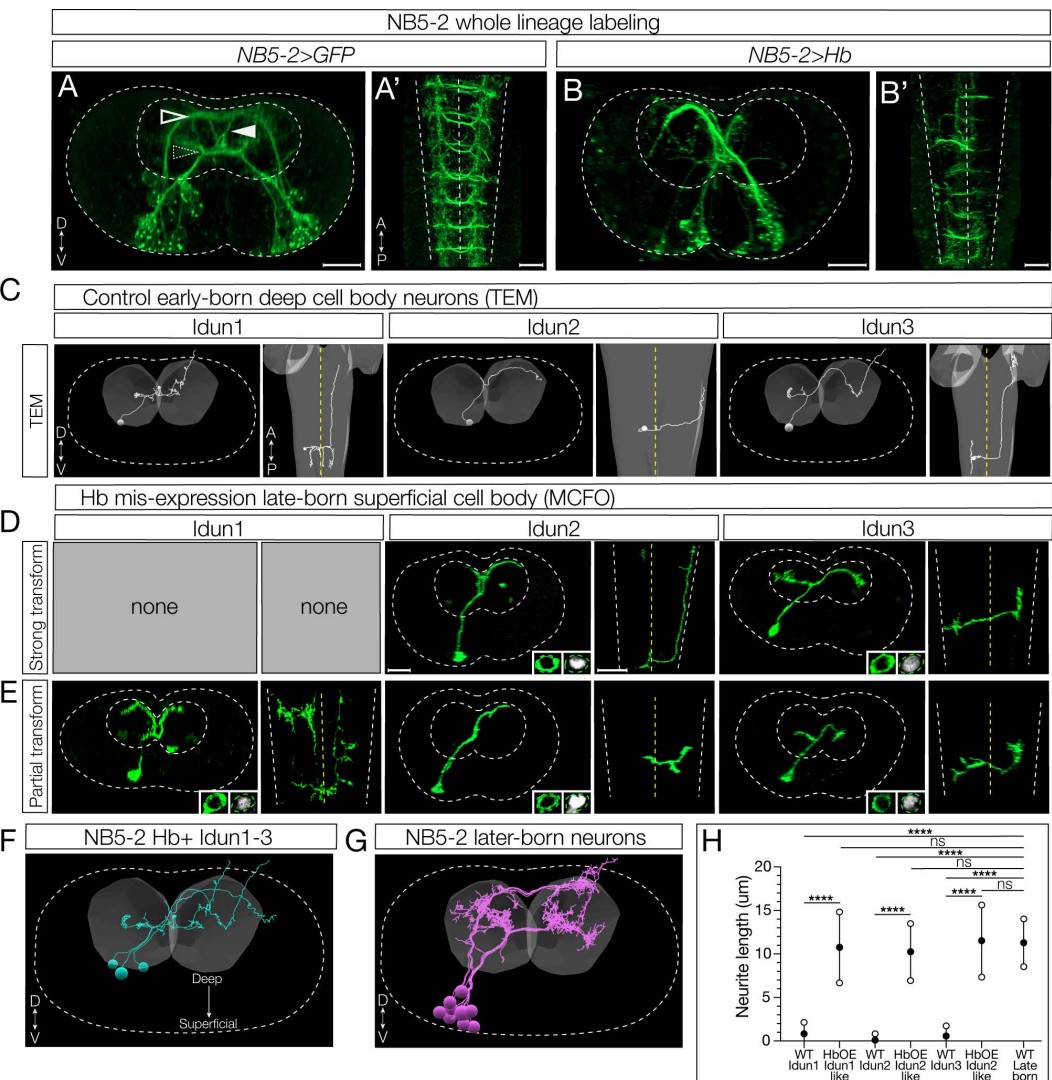

**Fig 7. Hunchback determines early-born neuron morphology.** (**A**) Whole lineage labeling of L1 (0−3 h ALH) wildtype (WT) NB5-2 > GFP (green) axon/dendrite morphology displaying dorsal (outlined arrow), ventral (dotted arrow), and diagonal (filled arrow) neurite projections within the A1/A2 VNC neuropil. (**A**) Dorsal up, posterior view; (**A′**) Anterior up, ventral view. Scale bar: 10 μm. (**B**) Whole lineage labeling of NB5-2 > Hb morphology within the A1/A2 VNC neuropil. (**B**) Dorsal up, posterior view; (**B′**) Anterior up, ventral view. (**C**) TEM reconstruction of WT Idun1 (left two panels), Idun2 (middle two panels), and Idun3 (right two panels) morphology and cell body position relative to the neuropil (gray). Left panel, dorsal up, posterior view; Right panel, anterior up, ventral view. VNC border (white dotted line); VNC midline (yellow dotted line). (**D**) L1 (0−3 h ALH) NB5-2 > Hb late-born neurons, labeled using MultiColor FlipOut (MCFO; green), displaying strong Idun1 (left two panels), Idun2 (middle two panels), and Idun3 (right two panels) morphology with Hb (right inset) expression in the cell body (left inset). Left panel, dorsal up, posterior view; Right panel, anterior up, ventral view. Scale bar: 10 μm. (**E**) NB5-2 > Hb late-born neurons displaying partial Idun1 (left two panels), Idun2 (middle two panels), and Idun3 (right two panels) morphology and Hb (right inset) expression in the cell body (left inset). (**F**) TEM reconstruction of WT NB5-2 Hb+ neuron morphology from a single A1 hemisegment. (**G**) TEM reconstruction of WT NB5-2 later-born neuron morphology from a single A1 hemisegment. (**H**) Quantification of WT Idun1−3, late-born transformed Idun1−3 NB5-2 > Hb (HbOE Idun1-like/Idun2-like/Idun3-like), and WT late-born neurite length (WT Idun1 avg = 0.83, $n = 8$; WT Idun2 avg = 0.11, $n = 7$; WT Idun3 avg = 0.58, $n = 5$; HbOE Idun1-like avg = 10.76, $n = 9$; HbOE Idun2-like avg = 10.25, $n = 10$; HbOE Idun3-like avg = 11.52, $n = 8$; WT late-born avg = 11.29, $n = 19$; one-way ANOVA *p-values*: WT Idun1 vs. HbOE Idun1-like *p-value* < 0.0001, WT Idun2 vs. HbOE Idun2-like *p-value* < 0.0001, WT Idun3 vs. HbOE Idun3-like *p-value* < 0.0001, WT Idun1 vs. WT late-born *p-value* < 0.0001, WT Idun2 vs. WT late-born *p-value* < 0.0001, WT Idun3 vs. WT late-born *p-value* < 0.0001, HbOE Idun1-like vs. WT late-born *p-value* = 0.99 (n.s), HbOE Idun2-like vs. WT late-born *p-value* = 0.97 (n.s), HbOE Idun3-like vs. WT late-born *p-value* > 0.99 (n.s). The data underlying the graphs in the figure can be found in S3 Data.

To prove that Hb misexpression reprograms late-born interneurons to generate an axon/dendrite morphology characteristic of early-born neurons, we labeled single neurons in the NB5-2 lineage using multicolored flip out (MCFO) [30] in a wild type or Hb misexpression background. Following Hb misexpression, we identified late-born interneurons based on their superficial cell body position (see Fig 3) [11], and confirmed that they were Hb+ despite their superficial position (Fig 7D and 7E). We found that these late-born neurons expressing Hb had an early-born axon/dendrite morphology, including a diagonal contralateral projection and ascending/descending intersegmental projections similar to the TEM reconstruction of Idun1-3 neurons (Fig 7C–7E); we call these "strongly" transformed neurons. In addition, we also observed "partially" transformed late-born neurons, characterized by the expected diagonal commissural projection but lacking ascending or descending projections (Fig 7E). Why there are two phenotypic classes is not clear (see Discussion).

To gain additional evidence for the Hb-mediated transformation of late-born neurons into neurons with morphological features characteristic of early-born neurons, we measured soma-neuropil distance as a proxy for birth order (see Fig 3) [11]. As expected, wild-type early-born Idun1, Idun2, and Idun3 neurons had cell bodies close to the neuropil, with very short soma-neuropil distance (Fig 7F, quantified in 7H). Also as expected, wild-type Hb-negative late-born neurons had superficial locations and a large soma-neuropil distance (Fig 7G, quantified in 7H). Importantly, Hb misexpression resulted in neurons with a diagonal commissural projection, characteristic of early-born neurons, yet showing a large soma-neuropil distance (Fig 7H). This shows that late-born, superficially located neurons can be efficiently transformed into neurons with early-born morphology following prolonged Hb expression. We conclude that Hb is sufficient to induce early-born interneuron morphology. This is consistent with the "TTF model" for specifying interneuron morphology.

## Hunchback determines interneuron presynapse targeting within the neuropil

We next wanted to determine if Hb played a role in early-born interneuron presynaptic targeting, which is particularly interesting due to the dense packing of synapses in the neuropil. We first sought to determine the presynapse localization of Idun1–3 within the neuropil, a critical first step in establishing proper connectivity to downstream early-born partners. We used the TEM dataset to predict the neuropil volume occupied by Idun1, Idun2, and Idun3 presynapses (Fig 8A), as well as for all NB5-2 neuronal presynapses (Fig 8B). The neuropil position of presynapses differed for each neuron, which we labeled as i1, i2, and i3 for Idun1–3, respectively. We then used Hb-filtered genetics (Fig 8C) to express Bruchpilot (Brp), an inactive presynaptic marker, in the Idun1–3 neurons and mapped the coordinates for the three volumes to the neuropil. Importantly, in controls, the Brp puncta targeted the same three volumes in both light and electron microscopy, which can be seen for both Hb+ neurons (Fig 8D) and all NB5-2 progeny neurons (Fig 8E). Note that left and right hemisegments were treated as independent data and mirrored to show only left hemisegments. Overexpression of Hb in the NB5-2 derived neurons results in several phenotypes. (1) There is a higher percentage of Brp puncta targeting i1 and i2 (compare Fig 8E and 5F; quantified in Fig 8G), consistent with late-born neurons being transformed to early-born Idun1 and Idun2 neurons, and targeting to their characteristic Idun1/2 neuropil volumes. (2) There is a loss of Brp puncta from late-born neurons (Fig 8H′ and 8I′), also consistent with late-born neurons being transformed to early-born Idun1 and Idun2 neurons. (3) There is a decrease in the percentage of Brp puncta that target the i3 volume. (4) There are Brp puncta abnormally positioned between the i2 and i3 volumes (Fig 8F; yellow arrows). Our results provide support for a model in which the early TTF Hb can reprogram late-born neurons, driving

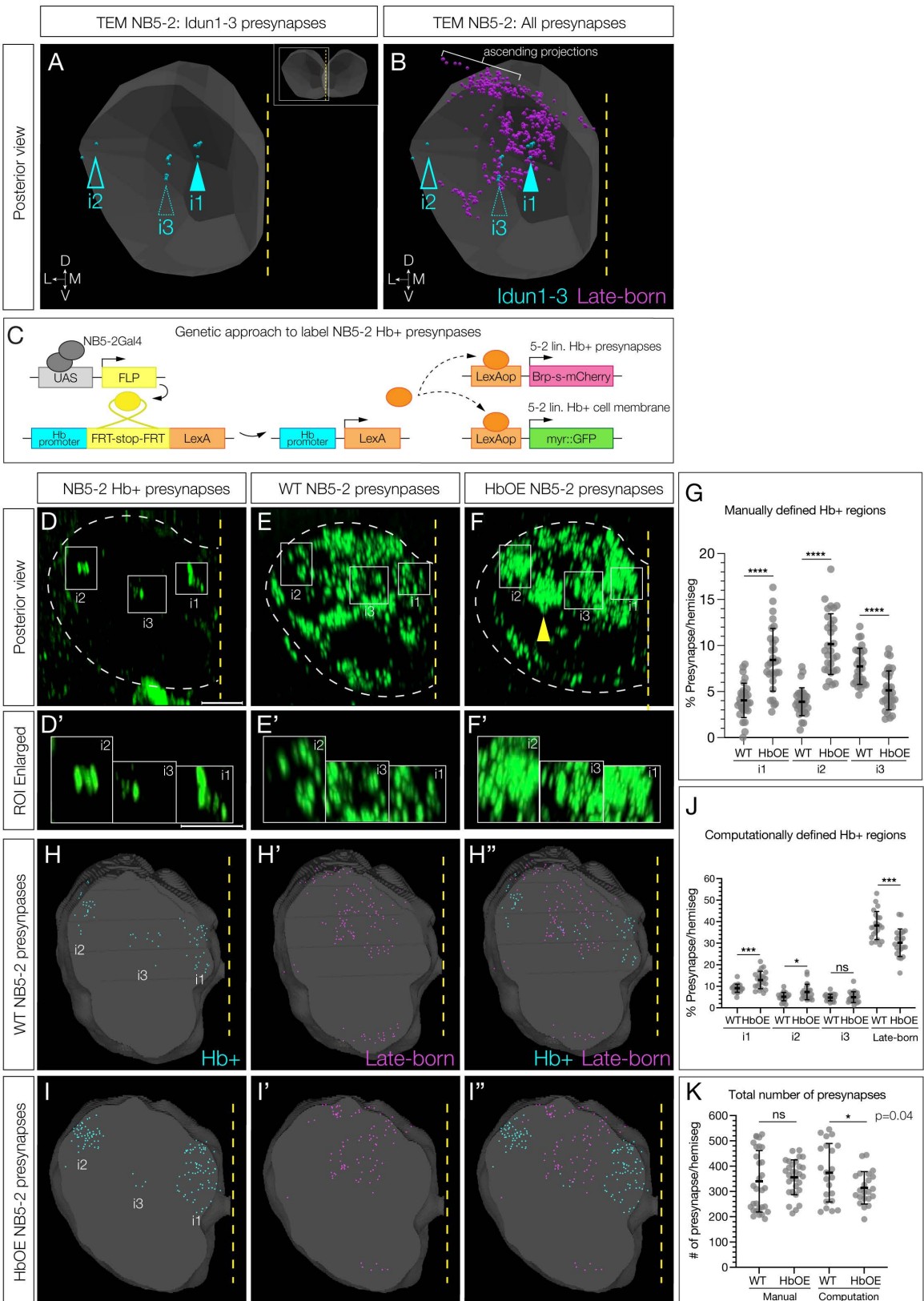

**Fig 8. Hunchback determines interneuron presynapse localization.** (**A**) TEM reconstruction of NB5-2 Hb+ Idun1 (i1), Idun2 (i2), and Idun3 (i3) presynapse positions in the A1 left hemisegment neuropil (gray). Dorsal up, posterior view. (**B**) TEM reconstruction of

NB5-2 Hb+ Idun1–3 (cyan) and late-born neuron (magenta) presynapses in the left A1 neuropil hemisegment. (**C**) NB5-2 Hb-filtered presynapse genetics. An intersectional genetics approach to drive expression of Brp exclusively in NB5-2 Hb expressing neurons. NB5-2 Gal4 (gray) drives expression of Flipase (FLP, yellow) to promote excision of a stop codon inserted downstream of the endogenous Hb promoter (cyan) and upstream of a LexA transgene (orange), allowing for LexA expression exclusively in NB5-2 Hb expressing neurons. LexA then drives expression of a truncated Brp conjugated to mCherry (Brp-s-mCherry, magenta) and GFP (myr::GFP, green). (**D**) NB5-2 Hb+ presynapses labeled by Brp expression (green) using intersectional genetics. Idun1–3 positions are outlined (white) in the left hemisegment neuropil (dotted outline) according to average location found from hemisegment centroid position (see Methods) Dorsal up, posterior view. Scale bar: 5 µm. Idun1–3 regions enlarged (**D′**). Scale bar: 3 µm. (**E**) WT NB5-2 presynapses labeled by endogenous Brp expression conjugated to a V5 epitope tag (green). Idun1–3 (i1–i3) positions are outlined; (**E′**) enlargement of boxed regions in E. (**F**) NB5-2 > Hb presynapses with Idun1–3 (i1–i3) positions outlined with a possible Idun3 mistargeting location (yellow arrow; see Discussion); (**F′**) enlargement of boxed regions in F. (**G**) Quantification of WT NB5-2 and NB5-2 > Hb (HbOE) larval presynapse percentage per hemisegment, found using manual registration, in neuropil positions i1 (WT avg = 4.05, HbOE avg = 8.45, $n$ = 30, *p-value* < 0.0001), i2 (WT avg = 3.90, HbOE avg = 10.16, $n$ = 30, *p-value* < 0.0001), and i3 (WT avg = 7.74, HbOE avg = 5.13, $n$ = 30, *p-value* < 0.0001). (**H**) WT NB5-2 presynapses in Hb+ (cyan) and late-born (**H′**, magenta) regions defined by computational registration. Merged image of Hb+ and late-born regions (**H″**). Images generated using Nepari (https://napari.org/). (**I**) HbOE NB5-2 presynapses in Hb+ (cyan) and late-born (**I′**, magenta) regions defined by computational registration. Merged image of Hb+ and late-born regions (**I″**). Images generated using Nepari (https://napari.org/). (**J**) Quantification of NB5-2 WT and HbOE larval presynapse percentage per hemisegment, found using computational registration, in neuropil positions i1 (WT avg = 9.11, HbOE avg = 12.93, $n$ = 22, *p-value* = 0.0002), i2 (WT avg = 5.19, HbOE avg = 7.35, $n$ = 22, *p-value* = 0.0161), i3 (WT avg = 4.79, HbOE avg = 4.98, $n$ = 22, *p-value* = 0.76), and late-born regions (WT avg = 38.31, HbOE avg = 30.23, $n$ = 22, *p-value* = 0.0002). (**K**) Quantification of all NB5-2 WT and HbOE larval presynapses per hemisegment found using manual and computation registration analyses (Manual WT avg = 340.2, Manual HbOE avg = 356, $n$ = 30, *p-value* = n.s.; Computation WT avg = 374, Computation HbOE avg = 314.2, $n$ = 22, *p-value* = 0.04). The data underlying the graphs in the figure can be found in S4 Data.

them to target their presynapses to neuropil volumes normally targeted by presynapses of early-born neurons.

To test these conclusions, we performed an additional analysis using custom Python code [31] to create a neuropil template, align our images to the template, and create a shared coordinate system across all images of all genotypes. As described above and in Fig 8D and 8D′, we mapped Hb+ NB5-2 lineage-derived presynapses into 3 Hb+ neuropil volumes (i1-i3); those outside these domains were presynapses of late-born NB5-2-derived neurons (S6 Fig). In controls, we observed presynapses in the three Hb+ domains (Fig 8H, cyan puncta) as well as in the late-born domains (Fig 8H, magenta puncta). In contrast, overexpression of Hb in the NB5-2 derived neurons resulted in a higher percentage of presynapses in two Hb+ neuropil domains (Fig 8I, quantified in 8J), similar to our manual registration findings (Fig 8D–8G). Importantly, we find no significant difference in the total number of presynapses between controls and Hb misexpression using our manual registration analysis and only a slight difference using our computational registration approach (quantified in Fig 8K). We conclude that Hb overexpression can drive late-born presynapses into Hb+ neuropil domains, based on both manual and computationally defined presynapse localization.

## Hunchback misexpression phenocopies loss of a proprioceptive behavior circuit

*Drosophila* larvae have a well-characterized proprioceptive circuit: proprioceptive sensory neurons> Jaam1-3> Eve + interneurons> Saaghi1-3> motor neurons [24,32]. Ablation of the Eve + interneurons in the proprioceptive circuit results in decreased larval crawling speed and increased C-shaped body bends [24]. Importantly, both Jaams and Saaghis are late-born neurons in the NB5-2 lineage [11]. This raises the question: if Hb misexpression transforms late-born Jaam and Saaghi neurons into an early-born identity, will they maintain or lose their connectivity to the proprioceptive circuit neurons? If Hb misexpression can drive late-born neurons into a circuit with endogenous early-born neurons, we would expect a loss of proprioceptive behavior. Thus, we misexpressed Hb throughout the NB5-2 lineage and

measured these two behaviors. Qualitatively, experimental larvae showed a strong uncoordinated crawling pattern (compare S1 and S2 Movies). Quantitatively, we observed significantly decreased forward locomotor velocity and increased C-shaped body bends, compared to controls (Fig 9A and 9B). We conclude that transformation of late-born NB5-2 interneurons to an early-born identity disrupts late-born proprioceptive circuit assembly or function.

## Discussion

During neurogenesis, intrinsic mechanisms play a vital role in determining neuron identity. The role of Hb in specifying motor neuron identity has been well studied in NB3-1 and NB7-1 [9,13,17,33–35]. Our study provides a comprehensive analysis of Hb in specifying NB5-2 interneuron identity. We found that the TTF cascade is expressed sequentially in NB5-2 (Hb > Kr > Pdm > Cas > Grh) and is transiently maintained in the post-mitotic progeny with single or multiple gene overlap combinations. This expression pattern is consistent with the previously characterized NB3-1 and NB7-1 and further support of a conserved, intrinsic TTF mechanism among VNC NBs [2]. To determine the role of Hb in specifying interneuron identity, we prolonged Hb expression in NB5-2 and analyzed three aspects of neuron identity: molecular identity, axon/dendrite morphology, and presynapse targeting (required for proper connectivity), and then assessed behavioral output due to changes in neuron identity.

We found that prolonged Hb expression prevented expression of late-born TTFs and increased the number of interneuron progeny with an early-born molecular identity (Nervy + Dbx + Nkx6+) at the expense of late-born progeny. In NB3-1, prolonged Hb expression increases the number of RP1/RP4 motor neurons with an early-born Zfh2-/Cut- molecular identity [12,16]. Similarly, NB7-1 Hb misexpression increases motor neuron progeny with an early-born U1/U2 molecular identity (Eve + /Zfh2−) [14,15]. In NB5-6 and NB7-4, both of which generate interneuron progeny, Hb binds NB-specific loci in each NB, suggesting differential expression of downstream genes [36]. Previous work has also shown that the late TTF Cas window of expression can be subdivided by "sub-temporal" genes to specify four distinct Apterous + (Ap+) interneurons in the NB5-6T lineage. Ap + neurons are uniquely specified through activation of the TF Collier/Knot (Col) and a series of feedforward loops [37,38]. The combinatorial expression of Kr/Pdm has also been shown to be crucial for the Nkx6 + molecular identity of the VO MN, derived from NB7-1 [18]. Thus, the combinatorial expression of

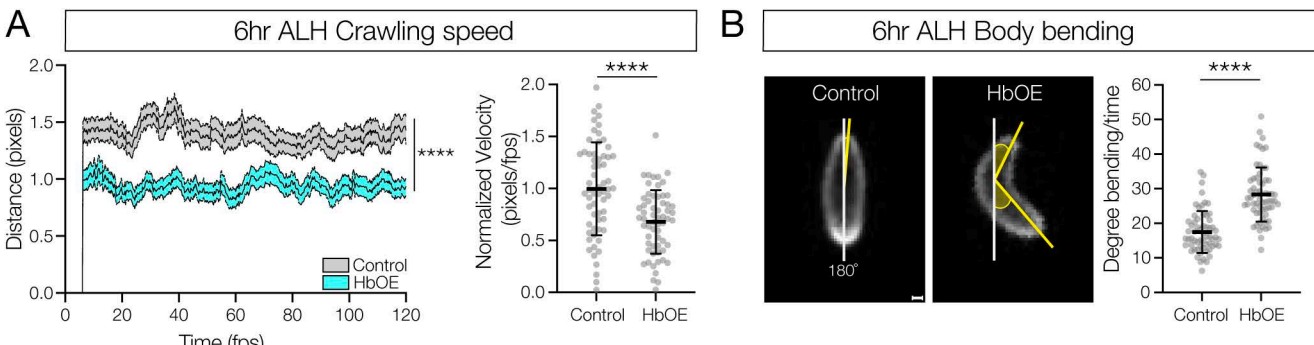

**Fig 9. Prolonged Hunchback expression disrupts late-born proprioceptive circuit connectivity.** (**A**) Crawling speed of newly hatched WT NB5-2 > GFP (Control; gray, avg AUC = 159, *n* = 61)) and NB5-2 > Hb (HbOE; cyan, avg AUC = 106.9, *n* = 62) larvae (0−4 h ALH) (*p-value* < 0.0001); Normalized crawling speed of WT (control, avg = 0.99) and HbOE (avg = 0.68) larvae (*p-value* < 0.0001). (**B**) WT NB5-2 (control, avg = 17.47, *n* = 61) and HbOE (avg = 28.31, *n* = 62) body bend posture determined by the angle of degree (yellow shaded) from 180° (white line); Quantification of body-bend degree from 180° (*p-value* < 0.0001). Scale bar: 40 μm. The data underlying the graphs in the figure can be found in S5 Data.

multiple TTFs may be important in promoting individual neuronal molecular identities. In the NB5-2 lineage, Hb-induced expression of the early-born TFs (Nervy, Dbx, and Nkx6) suggests that Hb may promote specification of distinct interneurons through differential expression of unique TF combinations.

To understand if Hb is required for proper NB5-2 interneuron morphology, we first identified the unique morphology of the NB5-2 Hb+ interneurons, Idun1–4, and then found that late-born neurons expressing Hb displayed a strikingly similar early-born diagonal neurite morphology. We found that late-born neurons displaying early-born morphology fell into two phenotypic classes: strongly transformed, where late-born neurons possessed a diagonal and ascending projection and partially transformed, in which the late-born neuron displayed a diagonal projection but lacked an ascending projection. We propose three explanations for these phenotypic classes: (1) partially transformed late-born neurons do not experience the same developmental environment as early-born neurons and thus lack the environmental signals to extend ascending and descending projections. (2) Late-born neurons inherently have less time to form long projecting neurites. Assaying morphology at later time points may allow for the formation of ascending projections. (3) Neurons may require a separate mechanism to form long projecting neurites, not present in some late-born neurons.

Prolonged Hb expression in NB7-1 results in a late-born to early-born transformation of U1/U2 early-born dendrite morphology [14,16]. The expression of extrinsic signaling ligand/ receptors, such as Sema/Plexin or Slit/Robo, allows a neuron to respond to a given attractant or repulsive signaling gradient within the neuropil [23]; differential expression of one or more receptor in early- or late-born neurons may determine differences in presynaptic or postsynaptic neuropil targeting. A transcriptomic study of *Drosophila* olfactory projection neurons (PNs) revealed that transcriptomes of PN subtypes show the highest amount of TF and cell-surface molecule diversity during circuit assembly [39]. In future studies, identifying and functionally testing downstream Hb targets such as guidance ligands/receptors will be an important next step in understanding how connectivity within the *Drosophila* CNS is established.

To understand if Hb promotes functional connectivity, we would ideally like to optogenetically simulate Idun1–3 presynaptic partners (or Idun1-3 directly) following Hb misexpression. A TTX-insensitive increase in GCaMP activity in putative downstream partners would functionally validate monosynaptic connectivity. Unfortunately, TEM connectivity data revealed that many Idun2 and Idun3 pre- and post-synaptic partners are unknown neurons, meaning we do not currently possess genetic access to these neurons. Idun1 shows pre- and post-synaptic connections to the well-characterized Moonwalking Descending Neuron (MDN) [40–43]. The connectivity between MDN and Idun1 is the most ideal connection among the NB5-2 Hb+ neurons because we have genetic access to MDN, and Idun1 expresses GABA (S5 Fig), suggesting it functions as an inhibitory neuron; however, there are two additional late-born NB5-2 neurons that also form synapses with MDN, making an excitation response between endogenous and transformed late-born neurons impossible to distinguish.

Although genetic limitations prevented us from testing whether Hb-transformed late-born neurons acquire functional connectivity matching that of early-born neurons, we performed two experiments that both support an Hb-induced switch in late-born neuron connectivity. First, we asked if Hb misexpression in late-born neurons induces them to form presynapses in neuropil domains normally targeted by endogenous early-born neurons. To achieve this, we use two complementary analyses of manual and computational registration to locate NB5–2 Hb+ neuropil subregions. We found a significant increase in the percentage of presynapses that target Idun1 and Idun2 subregions following Hb misexpression, consistent with the

transformed neurons acquiring the same connectivity as the endogenous early-born neurons. However, we also observed a significant decrease or no change in presynapses in the i3 volume and an abnormally positioned cluster of presynapses between i2 and i3 (Fig 8F; yellow arrows). These results may be consistent with Idun3 being transformed to Idun1 or Idun2; alternatively, the Idun3 presynapse region contains some late-born presynapses, and thus it is possible that these neurons are transformed into Idun1/2 identity. In the future, direct genetic assessment to Idun1–3 will allow us to explore these possibilities.

Second, we asked if Hb misexpression in late-born neurons disrupted their normal connectivity, leading to slow locomotion and increased C-shaped body bends, hallmarks of defective proprioceptive behavior [24]. Indeed, Hb misexpression resulted in the same behavioral defects as ablation of proprioceptive circuit neurons, consistent with reprograming late-born neuron connectivity to neurons of the proprioceptive circuit to a new connectivity that is unable to generate normal proprioceptive behavior. These findings are consistent with Hb misexpression abolishing normal late-born neuron connectivity. Note that the role of TTFs in establishing connectivity has been previously shown for embryonic motor neurons [14–16] and there are strong correlations between neuronal birth order and wiring specificity in *Drosophila* olfactory projection neurons and mushroom body Kenyon cells [44].

In *Caenorhabditis elegans*, the *hb* homolog, *hbl-1*, specifies early-born seam cell fate [45]. In addition, combinatorial expression of homeodomain proteins specifies specific neuron types [46]. Our data show that Hb is sufficient to promote the expression of two homeodomain proteins, Nkx6 and Dbx, in early-born NB5-2 progeny, and repress homeodomain Zfh2 expression. Hb repression of Zfh2 is also observed in early-born NB3-1 and NB7-1 neurons [12,14–16]. Taken together, this may place Hb as a hierarchical TF that promotes/represses the expression of downstream homeodomain proteins, which then generate unique, lineage-specific TF combinations to specify distinct aspects of individual neuron identity.

In both flies and mammals, neuron settling position is correlated with neuronal birth order. For example, early-born Hb+ neurons are located deep in the VNC cortex and late-born Cas+ are located superficially. In mammals, cortical projection neurons in different layers project to distinct regions of the CNS [47,48]. Progeny generated from aRG settle in a similar birth-order dependent manner, early-born neurons are located in deep cortical layers while later-born neurons migrate to settle more superficially [1]. The Hb mammalian ortholog, Ikaros (Ikzf1), is expressed early in aRGs and is necessary and sufficient to generate early-born progeny [6,7]. Prolonged Ikzf1 expression in aRGs increases the number of deep-layer early-born cells, identified by early-born TFs (Ctip2 + Tbr1 + Foxp2+) [7]. Similarly, Ikzf1 is expressed in early RPC and promotes the production of early-born neuron identities derived from RPCs. Misexpression of Ikzf1 in RPCs increases the number of early-born horizontal cells and amacrine cells at the expense of late-born bipolar neurons [6]. In addition, the Cas mammalian ortholog, Casz1, is expressed in later cell divisions of RPCs and promotes the production of late-born rod cells [49,50].

Our data show that Hb expression in NB5-2 is sufficient to specify early-born interneuron identity at the molecular, morphological, presynapse targeting, and behavioral levels. The correlations in *Drosophila* NBs and mammalian aRGs and RPCs suggest there is a conserved mechanism in which intrinsic TTF expression in progenitor cells is, in part, an initial requirement to diversify individual progenitor lineages. This leads to many open questions: how does Hb promote lineage-specific early-born neuron identity? What is the function of TFs downstream of Hb? Is the role of Hb in specifying interneuron identity conserved across multiple species? Does Ikzf1 promote/repress the expression of specific TFs important for distinct aspects of early-born cortical or retinal cell identity? These will be exciting avenues to explore in future studies.

## Materials and methods

### Fly stocks

Male and female *Drosophila melanogaster* were used. The chromosomes and insertion sites of transgenes (if known) are shown next to genotypes (Table 1).

*Gal4 lines, mutants, and reporters used were:*

 vnd-VP16 (2R, VK1, 59D3) [11]

 R16H05-Gal4 DBD (3L, P2, 68A4) (this work)

 10XUAS-IVS-myr::sfGFP-THS-10xUAS(FRT.stop)myr::smGdP-HA (RRID:BDSC_62127)

 10xUAS-IVS-myr::GFP (BDSC_32198)

 UAS-Hunchback (III) [9]

 hs-KD (BDSC_56167)

 3xUAS-FLPG5 (BDSC_55808)

 13xlexAop(KDRT-stop)SmGdp-Flag (BDSC_62111)

 hbP(FRT.myr::smGdP-cMyc.stop)LexA:p65-T2A-hb [11]

 hs-FLPG5.PEST.Opt (RRID:BDSC_77140)

 10xUAS(FRT.stop)myr::smGdP-OLLAS 10xUAS(FRT.stop)myr::smGdP-HA 10xUAS(FRT.stop)myr::smGdP-V5-THS-10xUAS(FRT.stop)myr::smGdP-FLAG (RRID:BDSC_64086)

 20xUAS-FLP (BDSC_55807)

 13xLexAop2-IVS-myr::sfGFP (this work)

 8xLexAop-Brp-short-mCherry [51]

 UAS-FLP (BDSC_4540)

 79C23S-GS(FRT-stop)smFPV5-2A-LexAVP16 [52]

### Hb ± filtered NB5-2 intersectional genetics

We used an intersectional genetics approach to label single NB5-2 Hb+ progeny (Fig 6A). Our NB5-2-Gal4 drove expression of UAS-Flipase (UAS-FLP) specifically in NB5-2, where it then bound to *Flipase Recombination Target* (FRT) *cis*-regulatory sites to promote excision of a stop codon downstream of the endogenous Hb promoter. Excision of the stop codon enabled LexA expression downstream of the *hb* promoter to be expressed following transcription of the endogenous *hb* gene. Stage 17 embryos were heat-shocked (see methods below) to promote expression of the KD recombinase [53] and excision of a stop codon downstream of the

**Table 1.** *Drosophila* genotypes utilized for the figures indicated.

| Genotype | Figure(s) |
|---|---|
| vnd-VP16; R16H05-Gal4 DBD/10xUAS-IVS-myr::sfGFP-THS-10xUAS(FRT.stop)myr::smGdP-HA | Figs 1–3 |
| vnd-VP16/10xUAS-IVS-myr::GFP; R16H05-Gal4 DBD | Figs 5, 7, and 9 |
| vnd-VP16/10xUAS-IVS-myr::GFP; R16H05-Gal4 DBD/UAS-Hunchback | Figs 4, 5, 7, and 9 |
| hs-KD, 3xUAS-FlpG5; vnd-VP16/ 13xlexAop(KDRT-stop)SmGdp-Flag; R16H05-Gal4 DBD/hb(FRT.stop)lexA-T2A-ORF | Fig 6 |
| 20xUAS-Flp; vnd-VP16/13xLexAop2-IVS-myr::sfGFP, 8xLexAop-Brp-short-mCherry; R16H05-Gal4 DBD/hbP(FRT.myr::smGdP-cMyc.stop) LexA:p65-T2A-hb | Figs 6 and 8 |
| hs-FLPG5.PEST; vnd-VP16; R16H05-Gal4 DBD/ 10xUAS(FRT.stop)myr::smGdP-OLLAS 10xUAS(FRT.stop)myr::smGdP-HA 10xUAS(FRT.stop) myr::smGdP-V5-THS-10xUAS(FRT.stop)myr::smGdP-FLAG | Fig 7 |
| hs-FLPG5.PEST; vnd-VP16/UAS-Hunchback; R16H05-Gal4 DBD/ 10xUAS(FRT.stop)myr::smGdP-OLLAS 10xUAS(FRT.stop)myr::smGdP-HA 10xUAS(FRT.stop)myr::smGdP-V5-THS-10xUAS(FRT.stop)myr::smGdP-FLAG | Fig 7 |
| vnd-VP16/UAS-FLP, 79C23S-GS(FRT-stop)smFPV5-2A-LexAVP16; R16H05-Gal4 DBD | Fig 8 |
| vnd-VP16/UAS-FLP, 79C23S-GS(FRT-stop)smFPV5-2A-LexAVP16; R16H05-Gal4 DBD/UAS-Hunchback | Fig 8 |

LexAop *cis*-regulatory site. The combination of FLP and KD excised stop codons promoted expression of spaghetti-monsterGFP conjugated with a FLAG epitope tag (smGdp-FLAG) specifically in NB5-2 derived Hb+ progeny.

Embryos were aged to stage 17 at 25°C for 18 h. Embryos were then heat-shocked in a 37°C water bath for 12 min, placed in an 18°C room for 12 min, and placed back at 25°C until larval hatching. First instar larval (L1; 0–3 h ALH) brains were then dissected and fixed following the fixation protocol (See Immunostaining and Imaging section).

Manual Presynapse Analysis (Fig 8C–8G)

NB5-2 Hb+ presynapses were labeled using a similar approach to the Hb filtered NB5-2 intersectional genetics (see above), to drive expression of a truncated, inactive version of the presynaptic marker, Bruchpilot (Brp), conjugated with an mCherry epitope tag (Brp-short-mCherry), and the cell membrane marker, myristoylated GFP (myr::GFP), in NB5-2 Hb+ progeny (Fig 8C), hereafter referred to as "Hb-filtered presynapses". Wild-type (WT) NB5-2 control and HbOE NB5-2 presynapses were labeled using the NB5–2-Gal4 driver and Synaptic Tagging with Recombination (STaR) [54].

L1 (0–3 h ALH) larval of each genotype (Hb-filtered presynapses, WT NB5-2 control, and HbOE NB5-2) brains were dissected and fixed (see Immunostaining and Imaging section). WT and HbOE presynapses were imaged using a laser intensity of 0.14 mW to capture all WT presynapses while also minimizing high fluorescent signal clusters observed in HbOE animals, to keep both in the lineage range.

Neuropil volume was labeled using N-Cadherin antibody staining. To define neuropil borders we made a surface object using Imaris 10.0.1 software. We then defined a centroid in each hemisegment by measuring the length, height, and depth of a single abdominal hemisegment, focusing our analysis on abdominal segments 1 and 2 (A1 and A2) (S7 Fig). For Hb+ neuropil volumes i1-i3, presynapses distribute along the anterior-to-posterior axis, spanning the depth of a single hemisegment. To control for segmental depth, we used the TF marker Even-skipped (Eve) to locate the U1 motor neurons and defined segmental borders according to their location (i.e., the U1 motor neuron position in thoracic segment 3 defined the A1 anterior border; the U1 position in A1 define the posterior border).

To isolate individual presynapses, we used the Imaris 10.0.1 spots tool, defining the size of an average presynapse spot for NB5-2 Hb-filtered (XY diameter: 0.31 μm, Z diameter: 1.71 μm) and WT and HbOE presynapses (XY diameter: 0.41 μm, Z diameter: 1.46 μm) and setting a minimum threshold intensity between 7 and 12AU. We then found the average center of each Hb+ position (i1 $n = 10$ animals, 20 hemisegments; i2 $n = 10$ animals, 20 hemisegments; i3 $n = 4$ animals; 9 hemisegments). To find the average distance of centroid to each center of the Hb+ regions, we measured and averaged the distances along the medial-lateral and dorsal-ventral axes from the centroid. A volume for each position was then created using Imaris 10.0.1 custom surface tool, spanning the depth of the hemisegment. The size of each volume was determined by the average center distance plus/minus 2 standard deviations. To validate this approach, we applied the averaged measurements of the i2 volume (the furthest Hb+ volume from the centroid) back to individual NB5-2 Hb-filtered presynapse hemisegments. We found that 5/6 hemisegments (3 animals) captured the labeled Hb+ presynapses in the i2 region. Total presynapse number per hemisegment was found using the Imaris 10.0.1 manual select tool and selecting all presynapse spots within a single hemisegment. Centroid position and Hb+ volumes for individual hemisegments were then found in WT control and HbOE hemisegments. Individual presynapses were isolated using the Imaris 10.0.1 spots tool (described above). Presynapses that fell within each Hb+ volume where then manually quantified. Presynapse percentages were calculated using the quantified presynapse numbers in Hb+ volumes and the total presynapse number in the corresponding hemisegment to minimize variance in Brp presynapse labeling between animals.

## Computational presynapse analysis (Fig 8H–8K)

To complement our manual analysis, we also compared confocal images from different animals on the same set of coordinates using the technique of template registration. Images of segments labeling NB5-2 Hb-filtered presynapses ($n = 2$ segments) and NB5-2 WT ($n = 4$ segments) and HbOE ($n = 4$ segments) presynapses were combined to make a "template" neuropil by overlaying and aligning segmented N-Cad stained neuropils using Computational Morphometry Toolkit Software (https://www.nitrc.org/projects/cmtk), Nepari (https://napari.org/), and custom Python code [31]. Individual images that successfully aligned with the "template" were selected for further analysis. Each hemisegment was analyzed separately by defining a transformation matrix that overlays left and right neuropils.

To define NB5-2 Hb+ presynapse regions within the neuropil, NB5-2 Hb+ presynapses from 17 A1/A2 hemisegments (8 animals) were isolated using the Imaris 10.0.1 spots tool and aligned to the "template" neuropil. After aligning to the hemisegment template, presynapse spots were blurred using a 1-μm Gaussian filter and thresholded so that 95% of the presynapses were included in the volume. Hb+ presynapse subregions (i1–i3) were then defined manually based on position on the medial-lateral axis.

Late-born NB5-2 presynapse regions were then defined by taking two randomly selected WT NB5-2 presynapse images and using the same Gaussian filter to create a volume where pixel intensity corresponds to nearby presence of WT NB5-2 presynapses (S6 Fig). Due to the difference in total presynapse number between Hb-filtered and WT presynapse images, Hb-filtered pixel intensity was scaled by a factor of 10 to equate pixel intensity to that in WT presynapse images. To exclude regions with many Hb+ presynapses, the WT presynapse volume was subtracted from the Hb-filtered volume. To quantify NB5-2 WT ($n = 22$ hemisegments, 5 animals) and HbOE ($n = 22$ hemisegments, 5 animals) presynapses, presynapse spots were defined using Imaris 10.0.1 spots tool for each genotype and aligned to the "template" neuropil. Presynapse spots that fell within the Hb+ and late-born regions in the left hemisegment were then quantified.

## Neuron naming

We named the early-born Hb+ interneurons in the NB5-2 lineage Idun1–4 after Iðunn, the Norse god of youth.

## Immunostaining and Imaging

Primary antibodies were chicken anti-GFP (1:1,000, RRID:AB_2307313, Aves Labs, Davis, CA), mouse anti-engrailed (5 μg/mL, DSHB 4D9-Eng, RRID: AB_528224), mouse anti-Hunchback (1:100, Abcam, F18-1G10.2), rabbit anti-Hunchback #5-27 (1:400) [12], guinea pig anti-Kr (1:500, Doe lab), rat anti-Pdm2 (1:100, abcam, ab201325, Cambridge, MA), rabbit anti-Castor (1:1,000, Doe lab), rat anti-Grainy head (1:400, a gift from Stefan Thor, University of Queensland), rat anti-Deadpan (1:20, abcam, ab195173), rabbit anti-Worniu (1:1,000; abcam, ab196362), rabbit anti-Nervy (1:300, gift from Richard Mann, Columbia University), Guinea pig anti-Dbx (1:200, Doe lab), rat anti-Nkx6 (1:500, gift from J. Skeath, Washington University, St. Louis), guinea pig anti-Runt (1:1,000, gift from C. Desplan, New York University), rat anti-Zfh2 (1:250, Doe lab), rat anti-FLAG (1:400, Novus NBP1-06712), rat anti-HA (1:1,000, MilliporeSigma, 11867423001, St. Louis, MO), chicken anti-V5 (1:800, Bethyl Laboratories, A190-118A, Centennial, CO), rabbit anti-mCherry (1:500, Novus, NBP2-25157), mouse anti-Eve (5 μg/mL, DSHB 2B8), rat anti-N-cadherin (0.168 μg/mL, DSHB DN-Ex #8), and fluorophores-conjugated secondary antibodies were from Jackson Immuno-noResearch (West Grove, PA) and were used at 1:300 for embryos and 1:400 for larval brains.

Embryos were fixed in 4% paraformaldehyde or formaldehyde for 20 min and stained as previously described [11]. Larval brains were dissected in PBS, fixed in 4% paraformaldehyde, and then stained by following protocols as described [55]. The samples were DPX mounted.

Images were captured with a Zeiss LSM 900 confocal microscope with a $z$-resolution of 0.22 μm. Due to the complex 3-dimensional pattern of each marker assayed, we could not show NB5-2 progeny marker expression in a maximum intensity projection, because irrelevant neurons in the $z$-axis obscured the neurons of interest; thus, NB5-2 progeny were montaged from their unique $z$-axis position while preserving their $X$–$Y$ position. Images were processed using Imaris (Bitplane, Zurich, Switzerland) to generate 3-dimensional reconstructions and level adjustments. Any level adjustment was applied to the entire image. Figures were assembled in Adobe Illustrator (Adobe, San Jose, CA).

## Behavior

We recorded behavior in newly hatched larvae (4–6 h). Behavior arenas were made with 1.2% agar, 2 mm thick. Arenas were placed on a functional independence measure (FIM) table [56] for recording. Larvae were transferred to the agar arena and given 2 min to acclimate before recording. Behavior was recoded for 2 min (125 fps) at 24 °C and 60% humidity using a Basler acA2040-25gm camera with a Computer TEC-V7X 1.1″ 7X Macro Zoom Telecentric lens and Pylon Viewer software. Recordings were edited in Fiji to eliminate arena borders, adjust the brightness-to-contrast ratio to increase larvae visibility (Min:15–19, Max:57–64), and saved at 25fps. Larval locomotion speed and body bends were calculated using FIMTrack 2.0 software.

## Statistics

Statistical significance is denoted by asterisks: ****$p < 0.0001$; ***$p < 0.001$; **$p < 0.01$; *$p < 0.05$; n.s., not significant. The following statistical tests were performed: Welch's $t$ test (normal distribution, non-parametric) (two-tailed $p$-value) (Figs 5Q–5X, 8G, 8J, 8K, 9A, 9B) and one-way ANOVA (Fig 7H). All analyses were performed using Prism8 (GraphPad). The results are stated as mean ± s.d., unless otherwise noted.

## Supporting information

**S1 Fig. NB5-2 can be identified by its stereotyped location in early embryos.** NB5-2 (yellow in schematic) was identified using the NB marker Dpn (magenta) and the row 6/7 expressing gene, Engrailed (En; green). NB5-2 was identified as the most medial Dpn+/En-negative NB, anteriorly adjacent to the En domain. NB5-2 shows Hb/Kr expression in early stage 9 embryos (top panels) and Pdm expression by early stage10 (bottom panels). Scale bar: 5 μm. (TIF)

**S2 Fig. Prolonged Hunchback expression in NB5-2 generates fewer progeny expressing late TTFs.** (A) Wildtype NB5-2 > GFP (green; top panels) and NB5-2 > Hb (bottom panels) progeny expressing Kr at stage 14. Anterior left, lateral view. Scale bar: 4 μm. (B) Wildtype NB5-2 > GFP (top panels) and NB5–2 > Hb (bottom panels) progeny expressing Pdm at stage 14. (C) Wildtype NB5–2 > GFP (top panels) and NB5–2 > Hb (bottom panels) progeny expressing Grh at stage 16–17. (D-H) Wild type and Hb overexpression quantified for the indicated markers; quantified in a minimum of 11 hemisegments from 3 embryos. The data underlying the graph in the figure can be found in S6 Data. (TIF)

**S3 Fig. Prolonged Hunchback expression increases the number of NB5-2 progeny that express Nkx6.** (A–B′) WT NB5-2 > GFP progeny (green, dotted outline) showing

co-expression of Hb and the early-born marker Nkx6 at stage 17. Anterior left, lateral view. Scale bars: 4 μm. (C–C′) NB5-2 > Hb progeny results in an increase in Nkx6 neurons.
(TIF)

**S4 Fig. NB5-2 late-born neurons do not possess a diagonal projecting morphology.** TEM reconstruction of wildtype NB5-2 late-born neurons organized by dorsal commissural projections (left column) and ventral commissural projections (right column). Dorsal up, posterior view (left panel); Anterior up, ventral view (right panel).
(TIF)

**S5 Fig. Idun1 expresses GABA neurotransmitter.** TEM reconstruction of Idun1 (upper panels) and single labeled Hb+ NB5-2 neuron genetically labeled with membrane-bound epitope tag, HA (green; bottom panels), with GABA expression (inset). Dorsal up, posterior view (left panels); Anterior up, ventral view (right panels). Scale bar: 10 μm.
(TIF)

**S6 Fig. Approach to defining late-born pre-synapse regions with computational analysis.** (A) Gaussian blur (1 μm) was applied to Hb+ NB5-2 presynapses (cyan; $n$ = 17 hemisegments, 8 animals) after aligning to a template neuropil and normalizing intensity by a factor of 10/hemisegment. A single optical slice is shown. Dorsal up, posterior view. (B) Gaussian blur (1 μm) was applied to randomly selected WT NB5-2 presynapse images (magenta; $n$ = 4 hemisegments, 2 animals) after aligning to a template neuropil and normalizing intensity by a factor of 1/hemisegment. Differing normalization factors between NB5-2 Hb+ and WT images were chosen to equalize intensities due to differences in presynapse number between NB5-2 Hb+ and WT images. (C) NB5-2 Hb+ regions (black) subtracted from WT NB5-2 presynapse regions (gray) to generate an image where pixel intensity corresponds to distance from late-born presynapses. (D) Pixels above a set threshold (red) define the late-born pre-synapse region. The pixel threshold was determined by setting the late-born template volume equal to the previously defined Hb+ presynapse volume.
(TIF)

**S7 Fig. Approach to identify NB5-2 Hb+ presynapse neuropil subregions.** Neuropil labeled with N-cadherin (white) to define the neuropil border (magenta dotted line; left panel). The neuropil border (magenta) was defined using Imaris 10.0.1 surface tool to find the centroid (cyan dot) of a hemisegment and the center of Idun1–3 presynapse position labeled with Brp staining (green; middle panel). Shown is the Idun2 presynapse neuropil position (i2; yellow dot). The average i2 coordinate location was then found by measuring the dorsal-ventral and medial-lateral distance from the centroid (yellow solid line; right panel). The size of the presynapse volume (white box) was determined by the average distance from the centroid ± 2 standard deviations. Individual presynapses were quantified using the Imaris 10.0.1 spots tool (gray dots). Dorsal up, posterior view. Scale bar: 5 μm.
(TIF)

**S1 Movie. Wild-type larval coordinated locomotion.** A representative L1 (0−3 h) wild-type larva (NB5-2 > GFP), free crawling on 1.2% agarose sheet. Video is accelerated 150% from real time.
(MP4)

**S2 Movie. Prolonged NB5-2 Hb expression showed strong uncoordinated crawling.** A representative L1 (0−3 h) NB5-2 > HbOE larva, free crawling on 1.2% agarose sheet. Video is accelerated 150% from real time.
(MP4)

**S1 Data.** Data underlying quantifications for Fig 2D and 2F.
(XLSX)

**S2 Data.** Data underlying quantifications for Fig 5Q–5X.
(XLSX)

**S3 Data.** Data underlying quantifications for Fig 7H.
(XLSX)

**S4 Data.** Data underlying quantifications for Fig 8G, 8J, and 8K.
(XLSX)

**S5 Data.** Data underlying quantifications for Fig 9A and 9B.
(XLSX)

**S6 Data.** Data underlying quantifications for **S2D**–**S2H Fig**.
(XLSX)

## Acknowledgments

We thank fellow lab member Noah Dillon for constructive comments on the manuscript. We also thank Tory Herman (Oregon) for comments on the manuscript. We thank lab members Sen-Lin Lai and Chundi Xu for reagents and the construction of fly genetics. We thank Stefan Thor, Richard Mann, Jim Skeath, and Claude Desplan for antibody reagents. We thank lab member Peter Newstein for writing the custom Python code and assistance in data analysis of presynapses shown in Fig 8H–8K. We thank lab member Kristen Lee for training and feedback on behavior experiments. Antibodies obtained from the Developmental Studies Hybridoma Bank, created by the NICHD of the NIH and maintained at the University of Iowa, Department of Biology, Iowa City, IA were used in this study. Stocks obtained from the Bloomington Drosophila Stock Center (NIH P40OD018537) and Vienna Drosophila Resource Center were used in this study.

## Author contributions

**Conceptualization:** Heather Q. Pollington.

**Data curation:** Heather Q. Pollington, Chris Q. Doe.

**Formal analysis:** Heather Q. Pollington, Chris Q. Doe.

**Funding acquisition:** Heather Q. Pollington, Chris Q. Doe.

**Investigation:** Heather Q. Pollington.

**Methodology:** Heather Q. Pollington.

**Project administration:** Chris Q. Doe.

**Supervision:** Chris Q. Doe.

**Visualization:** Heather Q. Pollington.

**Writing – original draft:** Heather Q. Pollington, Chris Q. Doe.

**Writing – review & editing:** Heather Q. Pollington, Chris Q. Doe.

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
