## [Editor Report · Decision Letter 0]

3 Oct 2024

Dear Dr Doe, 

Thank you for submitting your manuscript entitled "The Hunchback temporal transcription factor determines interneuron molecular identity, morphology, and presynapse targeting in the Drosophila NB5-2 lineage" for consideration as a Research Article by PLOS Biology.

Your manuscript has now been evaluated by the PLOS Biology editorial staff as well as by an academic editor with relevant expertise and I am writing to let you know that we would like to send your submission out for external peer review.

Once your full submission is complete, your paper will undergo a series of checks in preparation for peer review. After your manuscript has passed the checks it will be sent out for review. To provide the metadata for your submission, please Login to Editorial Manager (https://www.editorialmanager.com/pbiology) within two working days, i.e. by Oct 07 2024 11:59PM.

Kind regards,

Ines

--

Ines Alvarez-Garcia, PhD

Senior Editor

PLOS Biology

---

## [Decision Letter · Decision Letter 1]

22 Nov 2024

Dear Dr Doe,

Thank you for your patience while your manuscript entitled "The Hunchback temporal transcription factor determines interneuron molecular identity, morphology, and presynapse targeting in the Drosophila NB5-2 lineage" went through peer-review at PLOS Biology. Your manuscript has now been evaluated by the PLOS Biology editors, an Academic Editor with relevant expertise, and by four independent reviewers.

The reviews are attached below. As you will see, the reviewers find the results of your manuscript novel and interesting, but they also raise several issues that would need to be addressed. They make some suggestions for improvement, ask for missing information, and mention some points that should be discussed further.

In light of the reviews, we are pleased to offer you the opportunity to address the comments from the reviewers in a revision that we anticipate should not take you very long. We will then assess your revised manuscript and your response to the reviewers' comments with our Academic Editor aiming to avoid further rounds of peer-review, although might need to consult with the reviewers, depending on the nature of the revisions.

**IMPORTANT - SUBMITTING YOUR REVISION**

3. Resubmission Checklist

a) *PLOS Data Policy*

b) *Published Peer Review*

Sincerely,

Ines

--

Ines Alvarez-Garcia, PhD

Senior Editor

PLOS Biology

Reviewers' comments

Rev. 1:

The manuscript by Pollington and Doe on the role of Hunchback in determining interneuron identity in the Drosophila MB5-2 lineage is truly extraordinarily impressive. They provide detailed and thorough evidence that support the idea that the cascade of temporal transcription factors, rather than birth timing or order, for example, is responsible for the molecular identity, morphology, and presynapse targeting of interneurons in this lineage. Previous work from the Doe group has made this point in a compelling way for certain Drosophila motoneurons, but interneurons have been far more challenging to study due to the number and diversity of these cells, and the lack of markers to identify them individually. The experiments here report a truly fabulous amount of work at a uniquely high level of resolution. It is a testament to the previous work of this group and a relatively small cohort of other labs that the interpretation reported here will probably not come as a huge surprise in the field, but it is worth remembering that not that many years ago this whole notion would have been dismissed out of hand by many of those who were interested in the mechanisms underlying neuronal connectivity. That gives this work a great deal of significance in the larger context of developmental neuroscience.

I have few comments to make, and they are minor. I will confess that here and there one finds a couple of images where it is difficult for the non-expert to tell what distinguishes signal from nonspecific background in some cells of some micrographs, but I think this is quite unavoidable given the nature of the experiments. It does not cause me to question the quantification, or interpretations, of the authors. I also direct the authors to the paragraph of lines 398-405, where, unless I am mistaken, a sentence or two seem to be missing that would specify the phenotypic classes that are then discussed (though the point of the paragraph is clear and the caveats well-considered).

Rev. 2:

Pollington and Doe address the role of temporal transcriptional changes in neural stem cells on the differentiation of their progeny. They focused on three major identity features: molecular identity, axon/dendrite morphology, and presynaptic targeting. In conclusion, their study clearly shows that one can transform late born neurons into having identities of early born neurons by forcing the expression of an early factor in aged neural stem cells.

Similar studies have been published by this group and others on different neuronal lineages in Drosophila embryos. This study is unique in terms of its completeness. The authors first introduce the NB5-2 lineage: identify temporal factors expressed in the neural stem cells, identify transcription factors expressed in the progeny, and visualize morphology and synaptic output sites at single neuron level. In addition to this, they performed the functional analysis and demonstrated that they can force late-born neurons to adapt early-born cell fate identity by simply misexpressing Hb in neural stem cells.

Although the study is documented well in this manuscript, minor revisions needed.

* Figure 8 is not very convincing excluding the N2 region. The main reason is that the methodology is open for human errors at choosing the correct regions across different samples. Even within the same animal, left and right corresponding boxes don't match (e.g., N3 in Fig 8E). Depending on where you drove those boxes, you can get different results. Authors can keep the figure, but I suggest not making the conclusion for the N2/N3 regions.

* Authors need to indicate how many cells in average they see for their driver at stage 14? Do the authors see all the 18 expected cells (maybe a few lower because of apoptotic progeny) in their NB5-2-GAL > GFP labeling? If not, do they think this might cause issues related to their conclusions about the composition and birth order and how can they address this?

* Do we expect that the NB/GMC expression is inherited and maintained by both Notch OFF and OFF cell types? Is that the case in the previously established lineages? Or is it possible both cells inherit the GMC marker, one cell later represses it as seen in the Eve+ RP2 and its Eve- sibling cell?

* Conclusion made in Fig3 can be drawn from Fig 2. Do authors consider merging them into one figure?

* Figure 4 clearly shows that ectopic Hb expression can repress later TTs. However, it is not clear whether they checked Kr, Pdm expression across all time points. I assume they did, and they found the last 6 division occurs from Hb only NBs till stage 14. Do authors think that Hb expression after the competence window (stage 12-14) is able to reprogram the NB to Hb window? Or NB5-2 has a different competence? It would be good if these issues are addressed.

Minor issues:

* Figure 4 B WT Hb-Kr NB missing the Kr's blue color. It should be half/half but it is green only.

* Figure 7 : what stage embryos/ larvae used? If late-stage embryos used, this might explain the minor differences between light and EM morphology.

* Line 73-76: The author previously showed NB7-1 make a Nkx6+ Motor neuron in addition to U1-U5. It would be good to reiterate it here.

* Line 80-81: "early-born (late-born) muscle targets" implies muscles are born early but I am assuming author meant the neurons. Need to clarify.

* Materials: Dbx antibody source/dilution is missing.

* Line 124-126: Cited references did not show that NB5-1 lineage does not have Hb, Kr, Pdm expression. Considering it delaminates at S5, it is not expected to express these factors but It would be nice if the authors provide evidence for this or change the language to indicate it is expected.

* Line 168-169: It would be good to show the second Cas window for the NB5-2 in figure 1 by extending the time scale to 16hr embryos. It would be also nice to prior publication(s) showing the second Cas window.

* Line 251-254: The method section is missing the procedure to generate clones via hs-KD induction. When were the animals heat-shocked? during different stages of embryogenesis? Did they see over-represented clones of MN12/Idun 1 sibling cells assuming they hit the GMC?

* Figure 2C-C''': Why don't we see more Cas negative dorsal cells? From the figure legend it is not clear if shown cells are part of a larger clone.

* Is Fig 2E represent only the divisions until the end of stage 14?

* Line 766: "NB5-2>Hb progeny Hb (top panels)" not clear, is there typo here?

Rev. 3: Tzumin Lee – note that this reviewer has signed his review

This solid study by the Doe lab is well done. It reports in great details the VNC 5-2 lineage that yields distinct interneurons per temporal transcription factors. It establishes another model lineage for in-depth mechanistic studies from neuronal temporal fates to neuronal differentiation and ultimately organism behavior. The genetics to target and label the VNC 5-2 lineage with single neuron resolution is elegant. The phenotypes of TF expressions and neuron anatomy in wild-type and perturbed conditions are clearly described with beautiful figures. Discussions are faithful and insightful. I support publication of the paper as it is.

Rev. 4:

This manuscript presents a thorough analysis of the role that the temporal transcription factor Hb controls early fates in a lineage that had not been previously studied, and one that generates interneurons instead of motor neurons. For the most part, the genetic manipulations and data are unambiguous and in some cases elegantly designed. In particular, we appreciated the single neuron analysis and genetic labeling strategy used in Figures 6-8. There are a few questions that should be resolved before publication:

--There is ambiguity regarding the total number of Hb+ neurons. The text mentions 5 Hb+ neurons, whereas only 4 Hb+ neurons are visible in Figure 2A'.

--For Figures 1-4, please include the total number of replicates in each figure legend.

--In Figure 5, the data suggest that the total number of neurons remains constant despite Hb overexpression in late-born neurons. To verify this observation, performing BrdU staining is recommended.

--Using neuronal settling position as a proxy for neuronal birth order may lead to misidentification of a larval interneuron. Calculating the relative distance of newly born neurons with respect to their neuroblast (NB) may provide more precise information on each neuron's birth order.

--The authors use the TEM dataset to locate ROIs for the presynapse analysis, but how confident can they be that there is a direct 1:1 correspondence between the TEM and light microscope images? Given that the neurites are changing length, could the ROIs be shifting, leading to errors in the presynapse analysis?

--Do the 5-2>Hb flies hatch? What happens to this lineage post-embryonically?

---

## [Editor Report · Decision Letter 2]

22 Jan 2025

Dear Dr Doe,

Thank you for your patience while we considered your revised manuscript entitled "The Hunchback temporal transcription factor determines interneuron molecular identity, morphology, and presynapse targeting in the Drosophila NB5-2 lineage" for publication as a Research Article at PLOS Biology. This revised version of your manuscript has been evaluated by the PLOS Biology editors and the Academic Editor.

Based on our Academic Editor's assessment of your revision, we are likely to accept this manuscript for publication, provided you satisfactorily address the data and other policy-related requests stated below.

In addition, we would like you to consider a suggestion to improve the title:

"The Hunchback transcription factor determines interneuron molecular identity, morphology and presynapse targeting in the Drosophila NB5-2 lineage"

We expect to receive your revised manuscript within two weeks. 

*Published Peer Review History*

*Press*

Sincerely,

Ines

--

Ines Alvarez-Garcia, PhD

Senior Editor

PLOS Biology

Fig. 2D, F; Fig. 4Q-X; Fig. 7H; Fig. 8G, J, K; Fig. 8A, B; Fig. S2D-H and Fig. S3

Please also ensure that figure legends in your manuscript include information ON WHERE THE UNDERLYING DATA CAN BE FOUND, and ensure your supplemental data file/s has a legend.

---

## [Editor Report · Decision Letter 3]

19 Feb 2025

Dear Dr Doe,

Thank you for the submission of your revised Research Article entitled "The Hunchback transcription factor determines interneuron molecular identity, morphology and presynapse targeting in the Drosophila NB5-2 lineage" for publication in PLOS Biology. On behalf of my colleagues and the Academic Editor, Bing Ye, I am delighted to let you know that we can in principle accept your manuscript for publication, provided you address any remaining formatting and reporting issues. These will be detailed in an email you should receive within 2-3 business days from our colleagues in the journal operations team; no action is required from you until then. Please note that we will not be able to formally accept your manuscript and schedule it for publication until you have completed any requested changes.

PRESS

Sincerely, 

Ines

--

Ines Alvarez-Garcia, PhD

Senior Editor

PLOS Biology
